# Beyond Autoregression: Fast LLMs via Self-Distillation Through Time

**Justin Deschenaux, Caglar Gulcehre**
School of Computer and Communication Sciences
CLAIRE, EPFL
Lausanne, Switzerland
`{justin.deschenaux, caglar.gulcehre}@epfl.ch`

## Abstract

Autoregressive (AR) Large Language Models (LLMs) have demonstrated significant success across numerous tasks. However, the AR modeling paradigm presents certain limitations; for instance, contemporary autoregressive LLMs are trained to generate one token at a time, which can result in noticeable latency. Recent advances have indicated that search and repeated sampling can enhance performance in various applications, such as theorem proving, code generation, and alignment, by utilizing greater computational resources during inference. In this study, we demonstrate that diffusion language models are capable of generating at least 32 tokens simultaneously, while exceeding the performance of AR models in text quality and on the LAMBADA natural language understanding benchmark. This outcome is achieved through a novel distillation method for discrete diffusion models, which reduces the number of inference steps by a factor of 32-64. Practically, at the 1.3B parameters scale, diffusion models, even without caching, can generate tokens at a rate that is up to 8 times faster than AR models employing KV-caching, and we anticipate further improvements with the inclusion of caching. Moreover, we demonstrate the efficacy of our approach for diffusion language models with up to 860M parameters.

## 1 Introduction

In recent years, autoregressive (AR) large language models (LLM) have exceeded expectations (Vaswani et al., 2017; Devlin et al., 2018; Radford et al., 2019; Brown et al., 2020b; Kaplan et al., 2020; Raffel et al., 2020; Fedus et al., 2022; Hoffmann et al., 2022; Chowdhery et al., 2023; Google, 2023; Touvron et al., 2023). Importantly, many breakthroughs in coding (Chen et al., 2021), mathematics, and reasoning (Trinh et al., 2024b;a; Romera-Paredes et al., 2024; Hosseini et al., 2024; Wang et al., 2024) were achieved based on decoding large amounts of completions from a base LLM.

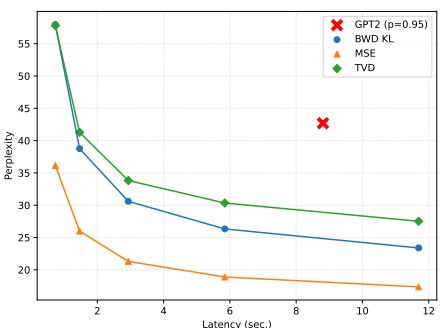

Figure 1: **Perplexity versus latency**. The diffusion models (169M) use 16, 32, 64, 128 and 256 decoding step.

Importantly, the benefits of repeated sampling can be so significant that it is often more efficient to use a smaller, faster model rather than a larger, slower one. More generally, one can improve the performance of a fixed model by scaling up computational resources at inference time (Madaan et al., 2023; Yao et al., 2023; Snell et al., 2024; Wu et al., 2024; Chen et al., 2024; Brown et al., 2024; Goyal et al., 2024), a phenomenon that was previously observed for games (Campbell et al., 2002; Silver et al., 2016; Lerer et al., 2019; Brown et al., 2020a; Jones, 2021). Hence, when tackling reasoning tasks, a major bottleneck is the latency of the model. In this work, we improve the decoding speed of LLMs by moving away from AR modeling. We build on recent breakthroughs in discrete diffusion (Lou et al., 2023; Sahoo et al., 2024; Shi et al., 2024; Ou et al., 2024). Our approach can generate text up to 8 times faster than AR models that

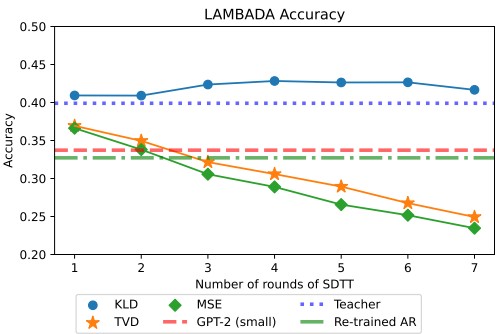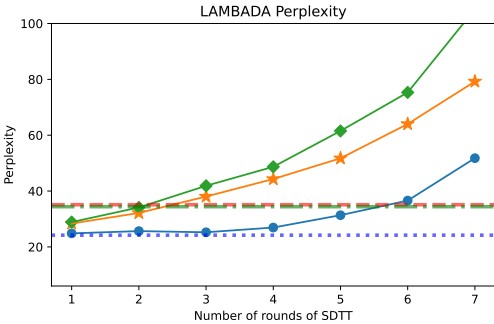

(a) Accuracy of the correct last word decoded from our model. Distillation with KLD loss leads the student model to outperform the teacher in terms of accuracy on LAMBADA.

(b) Perplexity of the last word. The KLD preserves performance best, and even when the student is trained to sample with 16 instead of 1024 steps, the student still matches AR baselines.

Figure 2: **Performance on LAMBADA after multiple rounds of SDTT** with different distillation losses. We pre-train with the masked diffusion language modeling objective (MDLM) (Sahoo et al., 2024) and distill with 7 rounds of SDTT. Note that a single word in the LAMBADA data set often consists of multiple tokens. We greedily decode all tokens a single forward pass for the diffusion models and decode autoregressively for the AR models.

use KV caching (Pope et al., 2022). Diffusion models are typically trained to maximize the evidence lower bound (ELBO), which does not consider the desired number of inference steps. Hence, vanilla diffusion models typically require thousands of decoding steps. Fortunately, it is possible to drastically reduce the inference costs of *continuous* diffusion models via distillation (Luhman & Luhman, 2021; Salimans & Ho, 2022). Continuous distillation methods rely on deterministic mappings from noise to data, such as DDIM (Song et al., 2022). The deterministic mappings can be efficiently learned by a student diffusion model to sample in fewer steps. We hypothesize that such deterministic map cannot exist for the diffusion language models studied in this work. Indeed, those models always initialize the denoising process with a sequence of masked token, hence a deterministic algorithm can only generate a single sample. As such, we devise a distillation method that does not does depend on deterministic maps. This is a significant finding because faster decoding mechanisms allow exploring a larger search space in applications that require search, planning, and reranking. In summary, our core contributions are as follows:

- We introduce *Self-Distillation Through Time* (SDTT), which allows generating **at least** 32 tokens at a time, while achieving better perplexity than GPT-2 with nucleus sampling for conditional and unconditional generation. Unlike many distillation methods for continuous diffusion models, SDTT does not rely on deterministic mappings such as DDIM (Song et al., 2022). SDTT is very simple and easy to implement.

- We show that SDTT can generate tokens up to 8 times faster than AR models that use KV caching, for models with 1.3B parameters, in 16 decoding steps. Importantly, the discrete diffusion model does not rely on activation caching, suggesting that there is potential for even greater efficiency gains. The latency gains for smaller models are even greater.

- We demonstrate the effectiveness of SDTT for models with up to 860M parameters. To the best of our knowledge, this represents the largest publicly available discrete diffusion language model.

- We evaluate the distilled students on LAMBADA (Paperno et al., 2016) and 6 multiple-choice questions benchmarks from Gao et al. (2021). We find that SDTT preserves the natural language understanding performance of the teacher.

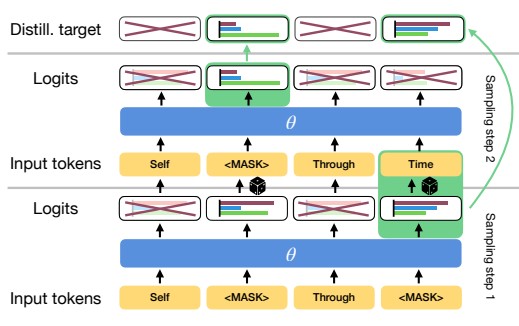 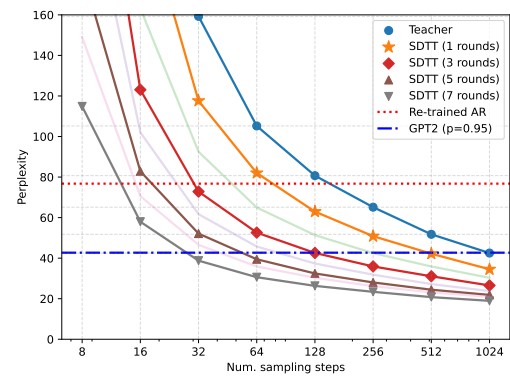

(a) The distillation targets are the log probabilities that lead to a token being denoised, concatenated with log probabilities of the last step for tokens that remain masked.

(b) SDTT on small models trained for 1M steps. Successive lines correspond to additional SDTT rounds. SDTT can outperform the teacher and GPT-2 with nucleus sampling.

Figure 3: **SDTT**. In figure (a), we illustrate how we prepare the distillation targets. In figure (b), we display the generative perplexity of samples after distillation.

## 2 BACKGROUND

### 2.1 MASKED DIFFUSION LANGUAGE MODELING

We follow the notation of Sahoo et al. (2024) to introduce masked diffusion language modeling (MDLM). Language modeling can be framed as the sequential prediction task of discrete tokens $(x_i)$ coming from a vocabulary $\mathcal{X} = \mathbb{Z}^{<N} = \{0, ..., N-1\}$ that can take $N$ possible discrete values. A language model would predict sequences of length $L$, which can be defined as the sequences of $x_i$'s originating from $\mathcal{X}^L = \left\{ \mathbf{x}^{(i)} = (x_0^{(i)}, \ldots, x_{L-1}^{(i)}) \right\}_{i \in \mathbb{Z}^{<K}}$. Let $\mathcal{D} := \left\{ \mathbf{x}^{(0)}, \ldots, \mathbf{x}^{(K-1)} : \mathbf{x}^{(i)} \in \mathcal{X}^L \right\}$ denote the training set. The goal of language modeling is to sample from the unknown distribution $p_0 : \mathcal{X}^L \to [0, 1]$ that generated the samples in $\mathcal{D}$.

Similarly to continuous diffusion, we sample from an approximation of $p_0$ by learning to denoise corrupted examples. One can sample from the model through ancestral sampling, starting from a stationary distribution. The stationary distribution of Sahoo et al. (2024) is such that all tokens of the sentence are replaced with a special MASK token like the MASK token used for pre-training BERT models. However, a key difference between BERT and MDLM is that MDLM is trained on sequences with varying levels of corruption, while BERT uses a fixed ratio.

**Discrete absorbing diffusion process** MDLM defines a forward process to corrupt data and a backward process to learn to recover data. MDLM uses a continuous-time formulation, with the data distribution denoted as $p_0$ and the stationary noise distribution as $p_1 = \boldsymbol{\pi}$. The forward process linearly interpolates between the one-hot distribution defined by the original document $\mathbf{x}$ and the stationary distribution $\boldsymbol{\pi}$, which places all mass on the MASK token. Mathematically,

$$q(\mathbf{z}_t|\mathbf{x}) := \text{Cat}(\mathbf{z}_t; \alpha_t \mathbf{x} + (1 - \alpha_t)\boldsymbol{\pi}), \tag{1}$$

where the noise injection schedule is defined by $\alpha_t$, for $t \in [0, 1]$. The constraints on $\alpha_t$ are that $\alpha_t \in [0, 1]$, $\alpha_t$ should be a strictly decreasing function of $t$, and $\alpha_0 \approx 1, \alpha_1 \approx 0$. The forward process is called absorbing because once a token is assigned to a MASK token, it cannot be reverted to a real token.

We can derive the analytical form of the reverse process $q(\mathbf{z}_s|\mathbf{z}_t, \mathbf{x})$, with $t > s$ and $\alpha_{t|s} = \frac{\alpha_t}{\alpha_s}$ as

$$q(\mathbf{z}_s|\mathbf{z}_t, \mathbf{x}) = \text{Cat}\left(\mathbf{z}_s; \frac{[\alpha_{t|s}\mathbf{z}_t + (1 - \alpha_{t|s})\mathbf{1}\boldsymbol{\pi}^\top \mathbf{z}_t] \odot [\alpha_s \mathbf{x} + (1 - \alpha_s)\boldsymbol{\pi}]}{\alpha_t \mathbf{z}_t^\top \mathbf{x} + (1 - \alpha_t)\mathbf{z}_t^\top \boldsymbol{\pi}}\right). \tag{2}$$

---

**Algorithm 1** Computing the *Self-Distillation Through Time* targets $\tilde{\mathbf{x}}_\theta^{\text{teacher}}(\mathbf{z}_t, t, m/k)$

---

1: **Inputs:** Noisy tensor $\mathbf{x}_t \in \mathbb{R}^{N \times L}$, Starting sampling time $t_{\text{start}} \in [0,1]^N$, Number of sampling steps $m/k \geq 2$, such that $m/k \in \mathbb{N}_+$, Sampling step size $\Delta \in (0,1)$, Mask token index $M \in \mathbb{N}$, Minimal sampling time $\epsilon$.
2: **Output:** Distillation targets $\tilde{\mathbf{x}}_\theta^{\text{teacher}}(\mathbf{z}_t, t, m/k)$

---

4: $\text{target} \leftarrow \text{zeros}(N, L, K)$       $\triangleright$ Allocate empty tensor for $\tilde{\mathbf{x}}_\theta^{\text{teacher}}(\mathbf{z}_t, t, m/k)$
5: $\mathbf{z} \leftarrow \mathbf{x}_t$
6: **for** $i = 0, ..., m/k - 1$ **do**
7:    $t_{\text{curr}} \leftarrow \max(t_{\text{start}} - i \cdot \Delta, \epsilon)$      $\triangleright$ Sampling step for the current time
8:    $\mathbf{z}_{\text{new}}, \ell_{\text{teacher}} \leftarrow \texttt{reverse\_sample}(\mathbf{z}, t_{\text{curr}}, \Delta)$    $\triangleright$ Updated $\mathbf{z}$ & log-probabilities $x_\theta(\mathbf{z}, t_{\text{curr}})$
9:    $U = \mathbf{z}_{\text{new}} \neq \mathbf{z}$       $\triangleright$ Create mask $U$ of tokens that were denoised
10:    $\text{target}[U] \leftarrow \ell_{\text{teacher}}[U]$      $\triangleright$ Extract log-probs for the denoised tokens
11:    $\mathbf{z} \leftarrow \mathbf{z}_{\text{new}}$        $\triangleright$ Update $\mathbf{z}$ for the next iteration
12: **end for**
13: $\text{target}[\mathbf{z} == M] = \ell_{\text{teacher}}[\mathbf{z} == M]$   $\triangleright$ Use log-probs of the last denoising step for masked tokens
14: **return** target        $\triangleright$ Target log-probs for all masked tokens in $\mathbf{x}_t$

---

**Objective and parameterization** To generate new samples, we can simulate the reverse process from eq. (2). Since the ground-truth sample $\mathbf{x}$ is unknown, Sahoo et al. (2024) learn an approximation $\mathbf{x}_\theta$ using a neural network with parameters $\theta$. Sahoo et al. (2024) then use $\mathbf{x}_\theta$ instead of $\mathbf{x}$ to simulate the reverse process. The sampling distribution is denoted as $p_\theta(\mathbf{z}_s|\mathbf{z}_t) := q(\mathbf{z}_s|\mathbf{z}_t, \mathbf{x}_\theta(\mathbf{z}_t, t))$. Sahoo et al. (2024) optimize $\theta$ using a continuous version of the negative evidence lower bound (NELBO) of Sohl-Dickstein et al. (2015a). Previous research has shown that continuous-time objectives optimize the data likelihood better (Kingma et al., 2023). Due to the definition of the absorbing diffusion process, the NELBO simplifies to a weighted cross-entropy loss between the ground-truth $\mathbf{x}$ and the model predictions $\mathbf{x}_\theta$:

$$\mathcal{L}_{\text{NELBO}}^\infty = \mathbb{E}_q \int_{t=0}^{t=1} \frac{\alpha_t'}{1 - \alpha_t} \log\langle \mathbf{x}_\theta(\mathbf{z}_t, t), \mathbf{x}\rangle \mathrm{d}t. \tag{3}$$

To derive eq. (3), Sahoo et al. (2024) impose two properties on $p_\theta(\mathbf{z}_s|\mathbf{z}_t)$. First, denoised tokens are never re-masked during sampling. Practically, this is achieved by manipulating the output of the neural network $\mathbf{x}_\theta(\mathbf{z}_t, t)$ to ensure that no probability mass is assigned to the MASK token. Secondly, already-denoised tokens are carried-over to the next sampling step. Sahoo et al. (2024) showed that both constraints lead to improved likelihood.

## 2.2 KNOWLEDGE DISTILLATION

Knowledge distillation (Bucila et al., 2006; Hinton et al., 2015) is a technique where a *student* neural network is trained to imitate the predictions of a more complex *teacher* model. One of the main advantages of distillation is the ability to reduce the inference cost associated with sampling from large LLMs while surpassing the performance of smaller models trained without distillation (Gu et al., 2024; Agarwal et al., 2024). The most relevant to our work are the distillation methods that match the predictions of the teacher and the student using a divergence measure $\delta$:

$$\mathbb{E}_{\mathbf{x}\sim\mathcal{D}}\left[\delta(\mu_s(\mathbf{x}_t|\mathbf{x}_{<t}); \mu_t(\mathbf{x}_t|\mathbf{x}_{<t}))\right], \tag{4}$$

Where $\mu_s, \mu_t$ are the AR distributions of the student and teacher, respectively, and $\mathcal{D}$ represent the training dataset. Common divergence measures include $f$-divergences (Wen et al., 2023) such as the Kullback-Leibler divergence (KLD) or the total variation distance (TVD).

---

**Algorithm 2** One training round of *Self-Distillation Through Time*

---

1: **Inputs:** Training set $\mathcal{D}$, Teacher $\mathbf{x}_\theta$, Divergence measure $\delta$, Number of sampling steps $m/k$,
   Sampling step size $\Delta$, Mask token index $M$, Total number of training steps $H$
2: **Output**: Distilled student $\mathbf{x}_\nu$.

---

3:
4: $\nu \leftarrow \theta$                                                    ▷ Initialize the student with the teacher weights
5: **for** $i = 0, ..., H - 1$ **do**
6:     $\mathbf{x}_0 \leftarrow \texttt{sample\_example}\,(\mathcal{D})$                              ▷ Sample a training example
7:     $t_{\text{start}} \sim \mathcal{U}[0, 1]$                                         ▷ Sample $t$ uniformly at random
8:     $\mathbf{x}_t \sim q_t(\mathbf{x}_t | \mathbf{x}_0)$                                    ▷ Forward diffusion process. See eq. (1)
9:     $\mathbf{x}_{\text{student}} \leftarrow \mathbf{x}_\nu(\mathbf{x}_t, t)$
10:    $\mathbf{x}_{\text{teacher}} \leftarrow \texttt{teacher\_SDTT}(\mathbf{x}_t, t_{\text{start}}, m/k, \Delta, M, \texttt{1e-5})$                ▷ See algorithm 1
11:    $\mathcal{L} \leftarrow \delta(\mathbf{x}_{\text{student}} || \mathbf{x}_{\text{teacher}})$        ▷ Compute divergence between student and SDTT targets.
12:    $\nu \leftarrow \texttt{backprop\_optim}(\mathcal{L}, \nu)$         ▷ Update the parameters of the student with AdamW
13: **end for**
14: **return** $\mathbf{x}_\nu$

---

## 3 METHOD

### 3.1 SELF-DISTILLATION THROUGH TIME

As explained in section 2.1, discrete diffusion language models optimize the NELBO over the training examples. Fewer decoding steps typically lead to lower sample quality because the approximation of the reverse process is less accurate, as visible in the teacher curve in fig. 4.

To address the issue of low sample quality with fewer decoding steps, we propose *Self-Distillation Through Time* (SDTT). SDTT fine-tunes a pre-trained MDLM to allow decoding with significantly fewer steps. Interestingly, our final model decodes samples with lower generative perplexity in 32 steps than the teacher would with 1024 forward passes. In short, SDTT improves the sampling speed by distilling the inference time computation to sample multiple steps into the student.

Let $p_\theta^{(m)}$ be the distribution of samples generated with $m$ steps, using a denoiser with parameters $\theta$. SDTT trains a denoiser with parameters $\nu$ to minimize a divergence $d$ between $p_\theta^{(m)}$ and $p_\nu^{(k)}$. Here $k < m$, and $k$ divides $m$ (e.g., $m = 1024$ and $k = 512$):

$$\min_\nu\ d\left(p_\nu^{(k)} || p_\theta^{(m)}\right). \tag{5}$$

Since $\mathbf{x}_\theta$ and $\mathbf{x}_\nu$ are the only learnable elements of the sampling process, they completely determine the sampling distributions $p_\theta^{(m)}$ and $p_\nu^{(k)}$. As such, training $\mathbf{x}_\nu$ to match the predictions of $\mathbf{x}_\theta$ with fewer steps minimizes eq. (5). We now present a method for generating targets $\tilde{\mathbf{x}}_\theta^{\text{teacher}}(\mathbf{z}_t, t, m/k)$ to train $\mathbf{x}_\nu$. Mathematically, we optimize the following objective:

$$\min_\nu\ \mathbb{E}_{\mathbf{z}_0 \sim \mathcal{D}, \mathbf{z}_t \sim q_t(\mathbf{z}_t | \mathbf{z}_0)}\left[\delta(\mathbf{x}_\nu(\mathbf{z}_t, t) || \tilde{\mathbf{x}}_\theta^{\text{teacher}}(\mathbf{z}_t, t, m/k))\right], \tag{6}$$

where $\delta$ a divergence measure between the student and the teacher targets $\tilde{\mathbf{x}}_\theta^{\text{teacher}}(\mathbf{z}_t, t, m/k))$. We consider the Kullback-Leibler divergence (KLD), Total Variation Distance (TVD), and Mean-Squared Error (MSE). See appendix B for details on those divergence measures.

**Generating the Teacher Targets** Following the terminology of knowledge distillation, we call the denoiser $\mathbf{x}_\theta$ used for many steps decoding as the *teacher* and the denoiser $\mathbf{x}_\nu$ used for a few steps decoding as the *student*. To train $\mathbf{x}_\nu$ to match the predictions of $\mathbf{x}_\theta$, we sample from the teacher for $m/k$ steps. Whenever a MASK token is denoised, we collect the log probabilities predicted by the teacher for this MASK token. These log-probabilities become the distillation targets $\tilde{\mathbf{x}}_\theta^{\text{teacher}}(\mathbf{z}_t, t, m/k)$. Algorithm 1 outlines this process and fig. 3a presents it visually. While fig. 3a shows how to distill two decoding steps in one, the procedure can be extended to larger values of $m/k$. The complete SDTT training loop is presented in algorithm 2.

**Iterated SDTT**    SDTT reduces the number of decoding steps by a factor $m/k$. If we want to reduce the number of decoding steps further, we can apply SDTT with $k' < k$, or alternatively apply SDTT $n$ times, using the newly distilled student as teacher for the next round, **which we refer to as iterated SDTT**. Instead of directly optimizing the divergence in eq. (5), we introduce $n$ intermediate distributions $p_{\nu_i}^{k_i}$ such that $m/k_i$ is an increasing sequence as a function of $i$. In practice, we choose $m = 2^{10}$ and $k_i = 2^{10-i}$ with $0 \le i \le 7$ and sequentially minimize the objective

$$\min_\nu \; d\left(p_{\nu_j+1}^{(k_{j+1})} || p_{\nu_j}^{(k_j)}\right), \tag{7}$$

for $0 \le j < 7$, where $\nu_j$ denotes the parameters of the $j$-th denoiser, with $\nu_0 = \theta$ (teacher). If the minimization procedure was perfect, minimizing eq. (5) or eq. (7) should result in the same solution. However in practice, we observe that it is easier to minimize eq. (7) sequentially for increasing values of $i$, in a progressive fashion, similar to Salimans & Ho (2022).

As an alternative to iterated SDTT, we tried using a single model and slowly growing the step size used to generate $\tilde{\mathbf{x}}_\theta^{\text{teacher}}(\mathbf{z}_t, t, m/k)$. Unfortunately, this approach was unstable and the loss diverged after 30-50 steps, irrespective of how small the sampling step size was. Similar behavior was observed by Norouzi et al. (2023).

## 4 EXPERIMENTS

We distill MDLMs on the OpenWebText dataset (Gokaslan & Cohen, 2019) as it was used to train recent discrete diffusion language models (Lou et al., 2023; Sahoo et al., 2024). We use the Adam optimizer with a learning rate of $6e-5$, a batch size of 128 and no weight decay. We linearly increase the learning rate for 500 training steps and keep it constant afterwards. As a base model, we reuse the checkpoint released by Sahoo et al. (2024). See appendix C for more details.

In section 4.1, we evaluate 3 distillation divergences and show that iterated SDTT can reduce the number of sampling steps by a factor 16-32. In section 4.2, we ablate on the importance of hyperparameters, including the duration of each round of iterated SDTT and the number of sampling steps to generate the targets $\tilde{\mathbf{x}}_\theta^{\text{teacher}}(\mathbf{z}_t, t, m/k)$. In section 4.3, we scale SDTT to models with of up to 860M parameters. Finally, in section 4.4, we compare the latency of SDTT against autoregressive models that use KV caching.

**Generative perplexity**    Following prior work (Dieleman et al., 2022; Lou et al., 2023; Sahoo et al., 2024), we use a larger model to compute the generative perplexity of unconditional and conditional samples. We evaluate the smallest students using GPT-2 (large) (Radford et al., 2019). In the scaling experiments, we use Llama3 8B (Touvron et al., 2023), since we compare models with up to 860M parameters. As noted by Zheng et al. (2024a), the generative perplexity is sensitive to the floating-point precision. In this section, we sample using bfloat16, and report results using float64 in appendix A. The conclusion are similar.

**MAUVE**    We evaluate conditional generation using the MAUVE score (Pillutla et al., 2021). MAUVE measures how well a model follows a prompt by comparing multiple generations with a reference continuation. We use the first 1024 samples with at least 1024 tokens from the WebText dataset (OpenAI, 2019), take the first 50 tokens as a prompt, and generate 50 tokens of continuation. For each prompt, we generate 5 continuations, as done in Lou et al. (2023).

**Sample diversity**    Post-training can drastically reduce the diversity of language models (Kirk et al., 2024; Agarwal et al., 2024; Li et al., 2024). Hence, we measure the diversity of samples using the self-BLEU score (Zhu et al., 2018) with the same completions used to compute MAUVE.

**Downstream performance**    We measure the downstream performance using the LAMBADA dataset (Paperno et al., 2016), as well as 6 multiple-choice question (MCQ) tasks from Gao et al. (2021). On LAMBADA, we report an upper bound on the perplexity, computed using the ELBO (3). We also report the suffix accuracy by masking all tokens of the last word and predicting all of them in a single forward pass, using the `argmax` of the predictions. The diffusion model is correct only if all the masked tokens are decoded correctly in a single decoding step. The 6 other benchmarks from Gao et al. (2021) evaluate the MCQ accuracy.

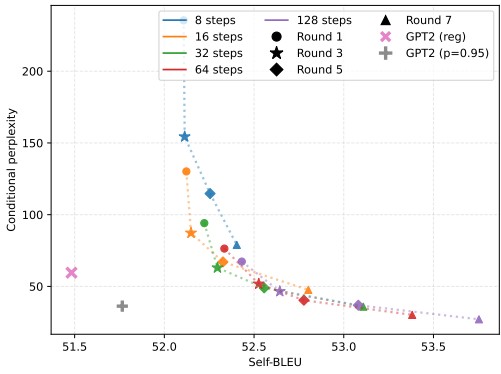 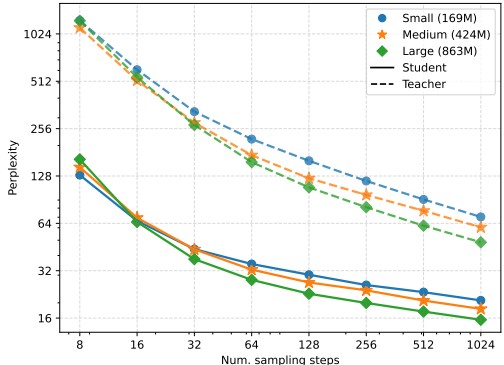

(a) **Diversity of conditional generation (small scale).** We measure the trade-off between quality and diversity using self-BLEU (Zhu et al., 2018). Deterministic sampling yields a score of 1. The diversity minimally decreases after distillation.

(b) **Scaling SDTT to 860M parameters.** The plot compares the performance of the teacher and **final** student (7 rounds). The student and teacher have the same size. The small distilled student reaches lower perplexity than the large teacher.

Figure 4: **Sampling step ablations on perplexity.** Perplexity of samples after each round of iterated SDTT. **(a)**: Iterated SDTT on a small model trained for 1M step. **(b)**: Scaling SDTT to larger models trained for 400K steps.

## 4.1 ABLATION ON THE TRAINING DIVERGENCE

SDTT requires choosing a divergence $\delta$ and we study the Mean-Squared Error (MSE), Total Variation Distance (TVD) and (reverse) Kullback-Leibler Divergence (KLD). We apply iterated SDTT for 7 rounds of $10k$ training iterations and generate $\tilde{\mathbf{x}}_\theta^{\text{teacher}}(\mathbf{z}_t, t, {}^m/k)$ with 2 sampling steps from the teacher (algorithm 1). We use an exponential moving average (EMA) of the weights with a decay of 0.9999 that we do not reset between rounds.

Figure 2 shows that students distilled with the KLD clearly outperform students trained using the MSE and TVD on LAMBADA. The LAMBADA accuracy of students tuned with the KLD slightly improves over the teacher, while the perplexity remains better or matches the AR baselines for all but the last round of SDTT. The improved accuracy on LAMBADA suggests that the model is better at predicting multiple tokens in parallel after distillation with SDTT, since we evaluates the accuracy by decoding all tokens of the last word simultaneously.

Figure 5 shows that the KLD seem to outperform the MSE and TVD objectives on MAUVE. Since we generate sequences of 100 tokens only for MAUVE, following (Lou et al., 2023), we sample with at most 128 steps, and use samples generated with 128 sampling steps from the teacher as a baseline. Note that as observed by Deschenaux & Gulcehre (2024), discrete diffusion models typically achieve slightly lower MAUVE scores than AR models. Nonetheless, distillation with the KLD objective improves the MAUVE score of the students. Similarly fig. 18 shows that continuations from the student distilled with the KLD reaches the lowest perplexity and match GPT-2 with nucleus sampling in 32 forward passes.

In table 1, we compare the downstream performance on the tasks of Gao et al. (2021) before and after distillation. We observe that SDTT minimally affects the results, and that student distilled with the KLD objective reaches higher accuracies than other students in all but one task

Figure 4a measures the diversity of samples using the self-BLEU score (Zhu et al., 2018), for the students distilled with the KLD objective. See appendix A for results with the MSE and TVD. We find that SDTT minimally decreases the diversity. Compared to distilling autoregressive models (Agarwal et al., 2024), SDTT minimally reduces the diversity. For reference, Agarwal et al. (2024) routinely observes an increase of 15 in self-BLEU while we observe a change of at most 2 for the KLD student. See appendix A for more results and details on the self-BLEU score.

Figure 6 shows that students distilled with KLD have higher *unconditional* generative perplexity than those distilled with the MSE. However, KLD is the only objective that preserves performance

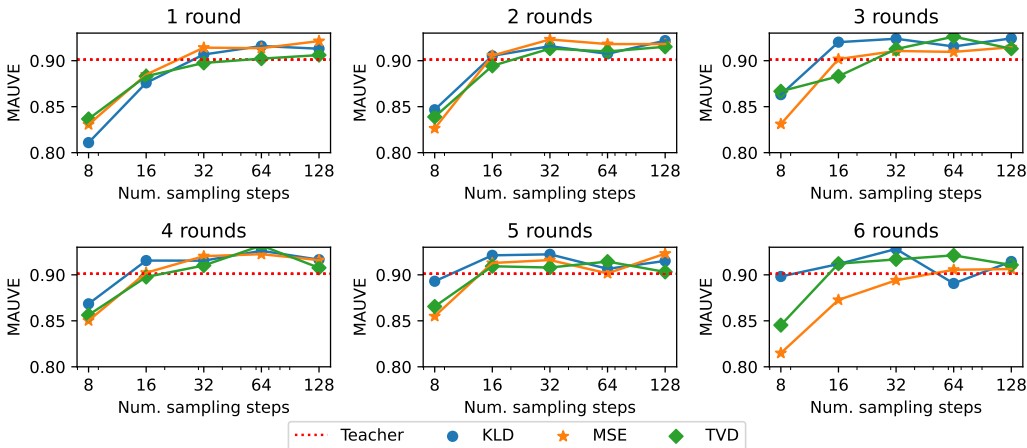

Figure 5: **MAUVE performance** of the student after each round of SDTT. The teacher performance is computed using samples generated with 128 decoding steps.

in the LAMBADA data set while still significantly reducing the generative perplexity compared to the teacher. Therefore, in the remainder of this work, we focus on the KLD.

## 4.2 Additional ablations

**Number of steps in each SDTT round** In section 4.1, each round of SDTT consists of $10k$ training iterations. Since the magnitude of the distillation loss does not reliably indicate convergence, we experiment with shorter rounds. We find that reducing the number of training iterations to $5k$ or $2.5k$ negatively impacted conditional generation performance, as shown in fig. 7. However, shorter rounds slightly improved the final generative perplexity (fig. 8) and resulted in marginally better LAMBADA perplexity (fig. 10). Since SDTT does not directly optimize the ELBO, an increase in perplexity is expected. Interestingly, the LAMBADA accuracy remains unchanged with shorter rounds.

**Number of sampling steps to generate the targets** In section 4.1, the targets $\tilde{\mathbf{x}}_\theta^{\text{teacher}}(\mathbf{z}_t, t, m/k)$ are generated using 2 sampling steps from the teacher. We explore distilling a larger number of sampling steps at once (4 or 8), since using more rounds of SDTT may induce more error accumulation in approximating the original teacher. Figure 13 shows that distilling more than two steps at a time is difficult and results in weaker results on LAMBADA. This suggests that the higher stochasticity of the targets generated with four or eight steps makes the task too difficult for the student.

**Generating targets with the analytical sampler** Lou et al. (2023) observe that using an analytical sampler (Campbell et al., 2022) results in higher quality samples compared to ancestral sampling. However, when generating targets $\tilde{\mathbf{x}}_\theta^{\text{teacher}}(\mathbf{z}_t, t, m/k)$ with analytical sampling, we observed minimal difference with ancestral sampling, as shown in fig. 11 and 12.

**Resetting the optimizer and Exponential Moving Average between rounds** Using an Exponential Moving Average (EMA) of the weights is known to improve the quality of samples from diffusion models (Nichol & Dhariwal, 2021). However, when applying SDTT for multiple rounds, it is unclear whether the EMA or current weights should be used as the teacher for successive rounds. Additionally, it could be favorable to reset the optimizer state between rounds as we grow the decoding step size. We experiment with two approaches: either resetting the optimizer state only, or resetting both the EMA and optimizer state. Figure 14 shows the generative perplexity when resetting the optimizer state and using the EMA as the teacher instead of the current weights, while fig. 15 presents the corresponding results for MAUVE. When using the EMA as teacher, since we accumulate updates in the EMA over 10k training iterations only, we use a slightly lower decay rate of 0.999. We find that using the EMA of the weights as the teacher may slightly improve performance.

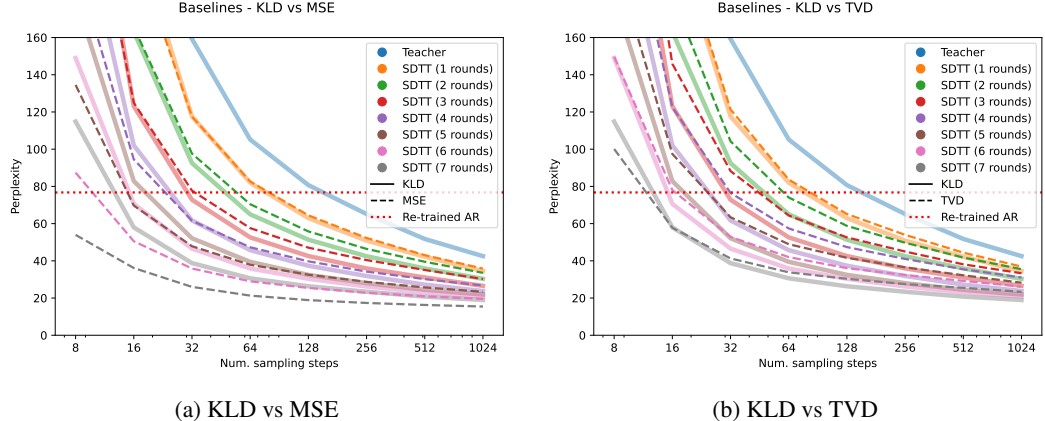

|  (a) KLD vs MSE | (b) KLD vs TVD |

Figure 6: **Perplexity for different losses and decoding step size**. Generative perplexity over 7 rounds of SDTT with MSE, TVD and KLD. While the KLD leads to a higher perplexity than the MSE; we focus on the KLD because it is the only divergence that retains the performance on the LAMBADA dataset.

### 4.3 SCALING SDTT TO 860M PARAMETERS

We apply SDTT to larger discrete diffusion models with up to 860M parameters. In this experiment, we train the models from scratch for 400k steps with a batch size of 512, a context length of 1024 and the Adam optimizer. We reuse the training configuration of Sahoo et al. (2024) and scale the models to larger sizes. We train 3 model sizes, small (169M), medium (424M) and large (863M). Details of the model architecture for each scale are shown in table 2. As for the other experiments, the models are diffusion transformers (Peebles & Xie, 2023) and we use an EMA with a decay of 0.9999. Although the results in section 4.2 suggest that short distillation rounds might be sufficient, it is unclear whether this result also holds on larger scales. Therefore, we use $10k$ steps per round of SDTT. For simplicity, we generate targets using 2 teacher ancestral decoding steps and do not reset the optimizer state or EMA between rounds.

Since we train larger models, we evaluate the generative perplexity using Llama3 8B (Touvron et al., 2023). The generative perplexity over the 3 model sizes is shown in fig. 4b. Interestingly, the smaller diffusion model (169M) sampled from with 64 steps or more after distillation achieves better generative perplexity than the largest model (863M) when sampling with 1024 steps. In fig. 16, we show that the MAUVE performance also improves after distillation for the medium and larger model. Finally, in fig. 17, we see that the LAMBADA accuracy improves after distillation, similar as in the smaller scale, when using the KLD objective.

### 4.4 LATENCY WITH SDTT

While SDTT allows sampling from discrete diffusion models with 32-64 times less decoding steps, a quantity of interest to practitioners is the actual latency of text generation. Indeed, while the reduction in the number of sampling steps is large, since discrete diffusion uses a non-causal architecture, we cannot use KV caching (Pope et al., 2022). KV caching improves the inference performance drastically for AR models, hence we compare the latency of SDTT with GPT-2 with KV caching. We successfully reproduce the results of Deschenaux & Gulcehre (2024), which showed a 4x improvement when sampling with 32 steps, and measure an 8x improvement with 16 decoding steps. We compute the latency using **untrained** models with around 1.3B parameters, using the same hyperparameters as Deschenaux & Gulcehre (2024). We use a batch size of 8 and time the sampling 10 times after one warm-up step on a single A100 GPU with 80 GiB of RAM. All models use FlashAttention (Dao et al., 2022). See Appendix A for additional experiments on the latency.

## 5    RELATED WORK

**Diffusion Models**    Diffusion models (Sohl-Dickstein et al., 2015b; Ho et al., 2020; Song & Ermon, 2020) are the basis of many state-of-the-art text-to-image models (Ramesh et al., 2022; Rombach et al., 2022; Saharia et al., 2022). After their introduction by Sohl-Dickstein et al. (2015b), Ho et al. (2020) showed that diffusion models can achieve FID scores (Heusel et al., 2017) comparable to GANs (Goodfellow et al., 2014; Arjovsky et al., 2017).

**Discrete Diffusion & Diffusion Language Models**    Prior to Sahoo et al. (2024); Shi et al. (2024); Ou et al. (2024), Lou et al. (2023) introduced a novel discrete diffusion language model called SEDD. When decoding with a large number of steps, SEDD can match or surpass GPT-2 in unconditional text generation. The model of Lou et al. (2023) learn a discrete generalization of the score of continuous diffusion models (Song & Ermon, 2020; Song et al., 2021). Campbell et al. (2022); Zhao et al. (2024) developed the continuous-time discrete diffusion framework. Hoogeboom et al. (2021) extended Bernoulli diffusion (Sohl-Dickstein et al., 2015b) to categorical distributions, and Austin et al. (2023) generalized the work of Hoogeboom et al. (2021) to more general corruption processes, including absorbing diffusion. Zheng et al. (2024b) develop a family of re-parameterized discrete diffusion models to enhance the training and decoding efficiency. In parallel, several studies have explored continuous diffusion for language modeling (Li et al., 2022; Dieleman et al., 2022; Han et al., 2023; Chen et al., 2023; Gulrajani & Hashimoto, 2024). Despite recent breakthroughs, diffusion language models still have some drawbacks (Deschenaux & Gulcehre, 2024). Ye et al. (2024) adapt Chain-of-Thought reasoning (Wei et al., 2023) to diffusion models.

**Distillation of Continuous Diffusion models**    Distilling continuous diffusion models is a well-studied area. For a comprehensive survey, see Luo (2023). Many distillation methods rely on Denoising Diffusion Implicit Models (DDIM) (Song et al., 2022), which showed that diffusion models can be sampled deterministically. Luhman & Luhman (2021) unroll trajectories sampled with DDIM and train a student to map noise directly to images. Luhman & Luhman (2021) pre-compute a dataset of noise-image pairs. Close to our work, Salimans & Ho (2022) teaches the student to match multiple sampling steps of the teacher, given corrupted training examples. However, unlike Salimans & Ho (2022), we cannot rely on the existence of a deterministic map via DDIM. Consistency distillation (Song et al., 2023) fine-tunes a pre-trained diffusion model to predict the final sample from intermediate points of the sampling trajectory, which enable faster sampling. Luo et al. (2024) distills a pre-trained diffusion model into single-step generator through a novel loss, *Integral Kullback-Leibler divergence*. *SD-XL Turbo* (Sauer et al., 2023) uses an adversarial formulation to sample with 1-4 steps from a latent diffusion model (Rombach et al., 2022).

**Masked & Non Auto-Regressive Language Modeling**    BERT (Devlin et al., 2018) introduced the masked language modeling objective. While BERT focuses on representation learning, discrete diffusion language models are generative. XLNet (Yang et al., 2020) uses a generalized AR pretrtaining method to model the text distribution over all permutations of the training sequences, outperforming BERT on downstream tasks. Pannatier et al. (2024) adopt a similar objective to XLNet for generative modeling instead of natural language understanding.

## 6    DISCUSSION

In this work, we introduce *Self-Distillation Through Time* (SDTT), a distillation method for discrete diffusion models. Recent works (Lou et al., 2023; Sahoo et al., 2024; Shi et al., 2024; Ou et al., 2024) suggest that discrete diffusion models can match or outperform autoregressive models in text quality. However, those models require more inference resources than AR models to achieve good performance, because of the non-causal architecture of the neural network that prevents the use of KV caching. We show that SDTT can reduce the number of decoding steps while retaining performance. Our final student is up to 8x faster than AR models that use KV caching and we demonstrate that SDTT is applicable to larger models as well. In future work, we plan to evaluate SDTT on tasks that involve generating a large number of completions from a base language model.

## 7 REPRODUCIBILITY STATEMENT

We provide details on model architectures, hyperparameters, and provide pseudocode for our algorithm. We built on top of the open source model of Sahoo et al. (2024), which makes it relatively easy for researchers to reproduce our results. Additionally, upon de-anonymization, we will release our code and artifacts.

## 8 ETHICS STATEMENT

Overall, language models are dual-use technologies, and thus, they can have unethical uses, such as fake content generation, and they can suffer from bias if applied to data sets that are not carefully curated. This paper focuses specifically on speeding up discrete diffusion language models at test time to reduce their computational demands; we do not have specific concerns with regard to this contribution.

## 9 ACKNOWLEDGEMENTS

We thank the ICLR'25 reviewers, area chairs, and organizers for their valuable feedback and support. We acknowledge the SCITAS team at EPFL for providing access to their beta cluster, and Karin Gétaz for her administrative assistance. This work was supported by the Swiss AI Initiative through a grant from the Swiss National Supercomputing Centre (CSCS), project ID a10 on Alps. Special thanks to Skander Moalla for providing a reproducible compute infrastructure code template.

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

Table 1: **Downstream evaluation results**. We report the accuracy of GPT-2, the teacher and students after 7 rounds of SDTT. Distillation seems to minimally affect the downstream performance.

| Task | GPT-2 | Teacher | KLD student | MSE student | TVD student |
|------|-------|---------|-------------|-------------|-------------|
| ARC-Easy | **43.81** | 40.91 | 40.57 | 40.45 | 40.32 |
| ARC-Challenge | 19.03 | **21.08** | 20.73 | 19.28 | 20.05 |
| HellaSwag | 28.92 | **30.50** | 29.65 | 29.10 | 29.18 |
| MathQA | 21.21 | 21.78 | 21.47 | **22.28** | 21.84 |
| PIQA | **62.89** | 59.74 | 59.85 | 58.11 | 58.16 |
| WinoGrande | **51.62** | 50.91 | 50.75 | 49.57 | 50.36 |

## A    ADDITIONAL ABLATION RESULTS

In this section, we show additional plots on the ablations we conducted. Because the KLD was best in retaining the performance on the LAMBADA dataset, we used it in most the ablations. Hence, unless specified, the following experiments distill using the KLD.

**Generative perplexity and precision of the floating-point operations.**    Zheng et al. (2024a) observed that low-precision sampling can be problematic in masked diffusion models, leading to reduced diversity and potentially misleading generative perplexity scores. As such, in addition to bfloat16, we try distilling (i.e. computing the backward KL) and sampling using 64 bits precision. Overall, it does lead to a higher generative perplexity, however the conclusions remain similar, as the final student achieves lower generative perplexity than GPT-2 with nucleus sampling (p=0.95) in 64 sampling steps, as shown in fig. 9.

**Ablations on the number of steps per round of SDTT**    In fig. 7 we show the MAUVE performance. In fig. 8 we show the generative perplexity, and in fig. 10, we show results on LAMBADA.

**Ablation on the analytic sampler**    In fig. 11 we show results on LAMBADA, and on fig. 12 the MAUVE score.

**Distilling more than 2 steps at once**    In fig. 13, we show the generative perplexity.

**Ablation on the optimizer state and exponential moving average of the weights**    In fig. 14 we show the generative perplexity when resetting the EMA and optimizer state. In fig. 14, we compare the generative perplexity when resetting the optimizer state only, and when resetting the EMA state. Finally, in fig. 15, we show the MAUVE score.

**Plots for scaled SDTT**    In fig. 16 we show the MAUVE score and in fig. 17, we show results on LAMBADA.

**Conditional perplexity with TVD**    In fig. 18c, we show the conditional perplexity (prompt excluded) on the small scale, for models trained for 1M steps. Empirically, the TVD performs worse than the KLD and MSE.

**Measuring the diversity**    We evaluate the generation diversity using the self-BLEU score (Zhu et al., 2018). The self-BLEU score averages the BLEU score between one completion and the others. Therefore, when the sampling algorithm is deterministic, the self-BLEU score is 1, and a lower self-BLEU score denotes a more diverse set of samples. Formally, let $X = \{x_1, ..., x_n\}$ be conditionally-generated sequences, starting with the same prompt. The self-BLEU score can be computed as

$$\text{self-BLEU} := \frac{1}{n} \sum_i \text{BLEU}(x_i, X \setminus \{x_i\}). \tag{8}$$

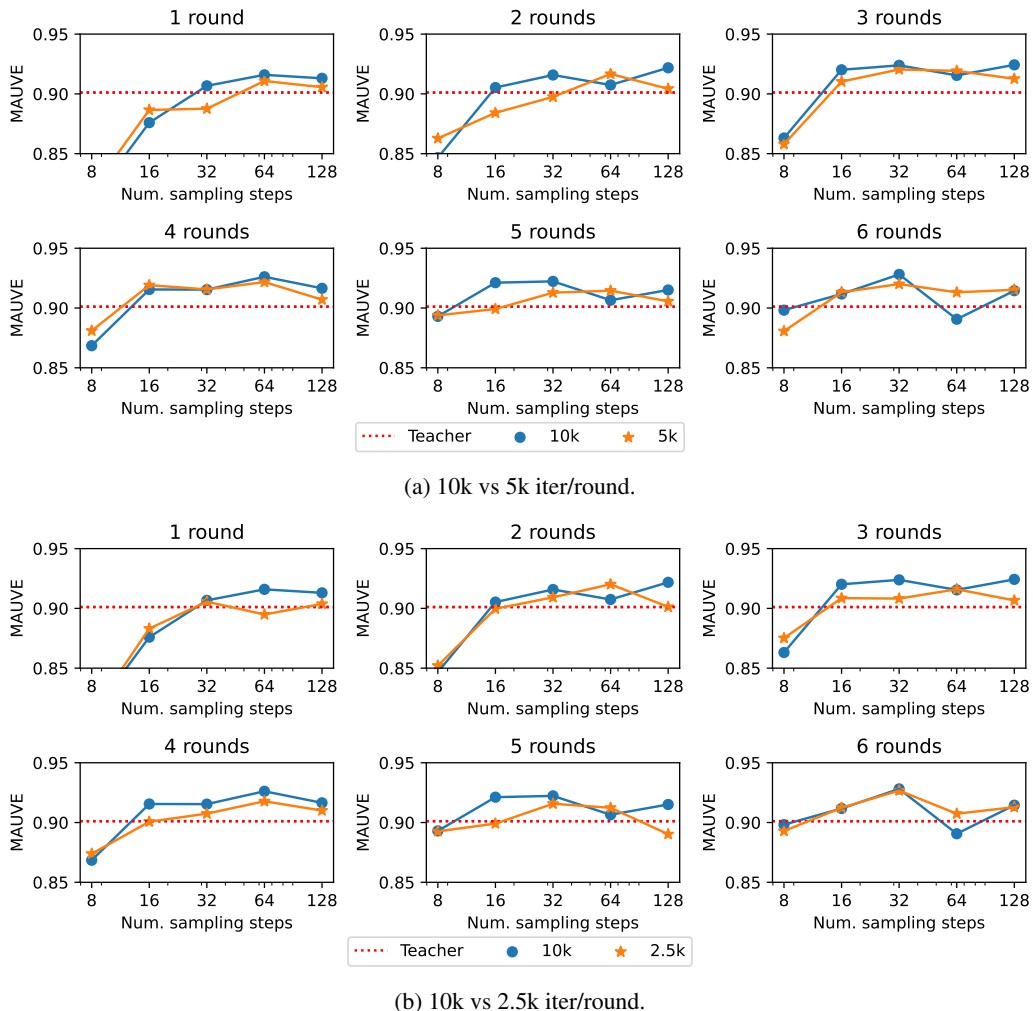

(a) 10k vs 5k iter/round.

(b) 10k vs 2.5k iter/round.

Figure 7: MAUVE performance with fewer steps per distillation round. It seems that using 5k or 2.5k distillation steps instead of 10k per round is detrimental to the MAUVE performance.

We compute the self-BLEU score using 1000 prompts, as for MAUVE, and generate 5 continuations per prompt. Figure 4a, fig. 19a and fig. 19b show the self-bleu score after distillation with the KLD, MSE and TVD objectives. Each objective only minimally decrease the diversity after distillation. Compared to on-policy distillation of autoregressive models (Agarwal et al., 2024), the decrease is marginal, as Agarwal et al. (2024) observe an increase of self-BLEU of the order of 10-20, demonstrating a more significant decrease in diversity.

**Decoding latency** In addition to the results on the 1.3B scale, we report the latency for models with 169M, 424M, 863M, 3B and 8B parameters. We compute the latency with a batch size of 8 and 4. Figure 20 shows the latency with a batch size of 8 and fig. 21 using a batch size of 4. Figure 22 shows the trade-off between latency and perplexity. We measure the latency at the small model size and compare GPT-2 with the final students after 7 rounds of distillation.

**Additional downstream evaluation results** We show the performance of GPT-2, the teacher and distilled students on additional downstream benchmarks from Gao et al. (2021) in table 1.

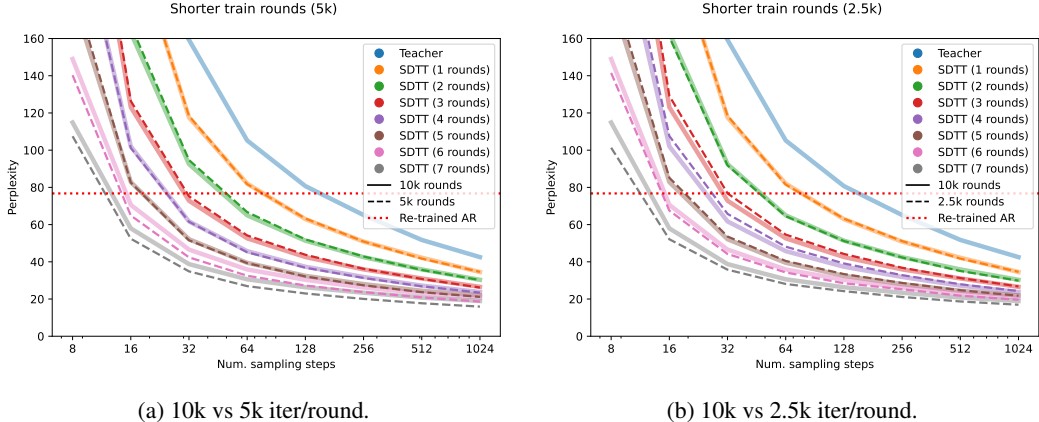

(a) 10k vs 5k iter/round.

(b) 10k vs 2.5k iter/round.

Figure 8: Generative perplexity with fewer steps per distillation round. Using 5k or 2.5k steps per round yields slightly improved perplexity after the latest distillation rounds while being a slightly worse in intermediate ones.

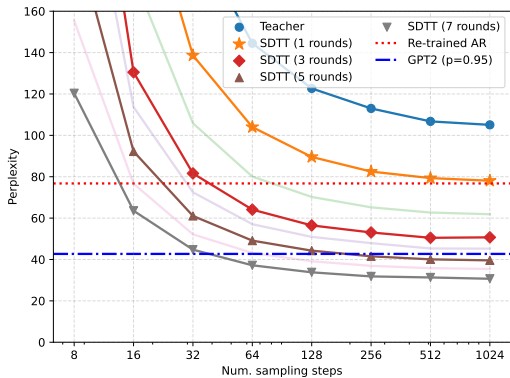

Figure 9: Generative perplexity when distilling and sampling with 64 bits precision. Namely, we sample from the teacher and students in float64, and compute the backward KL in float64.

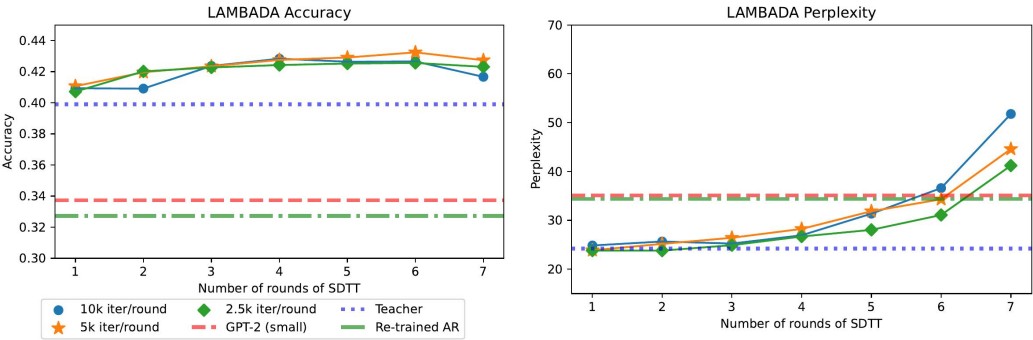

Figure 10: Performance on LAMBADA when distilling with fewer steps per distillation round.

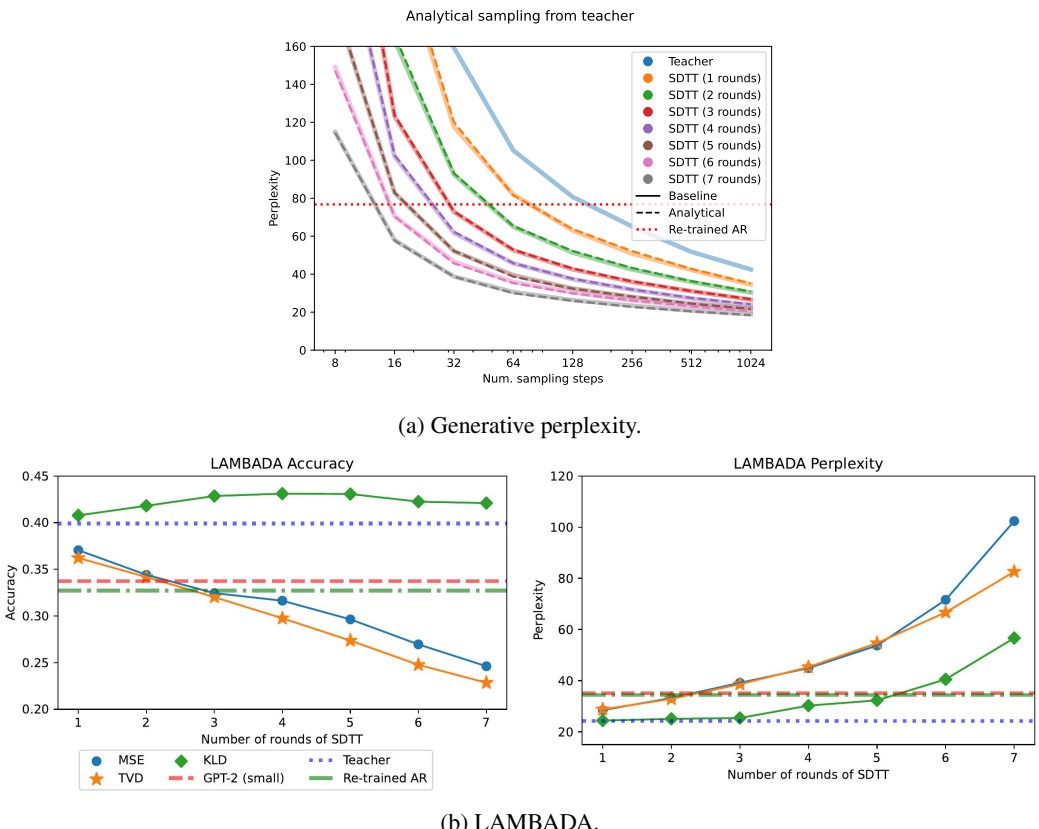

(a) Generative perplexity.

(b) LAMBADA.

Figure 11: Generative perplexity and performance on the LAMBADA dataset when using the analytical sampler. We find no clear benefit over the ancestral sampler.

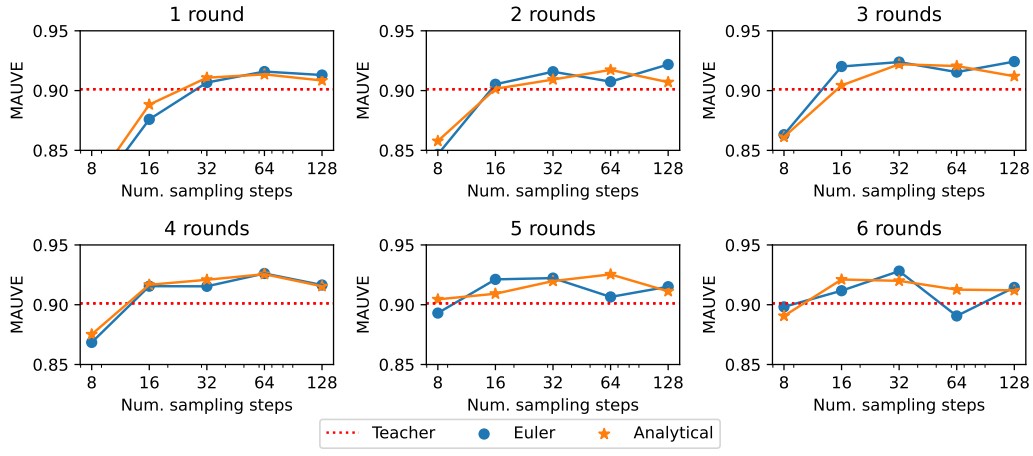

Figure 12: MAUVE performance when distilling using the ancestral sampler used by Lou et al. (2023). We find no clear benefit over the ancestral sampler.

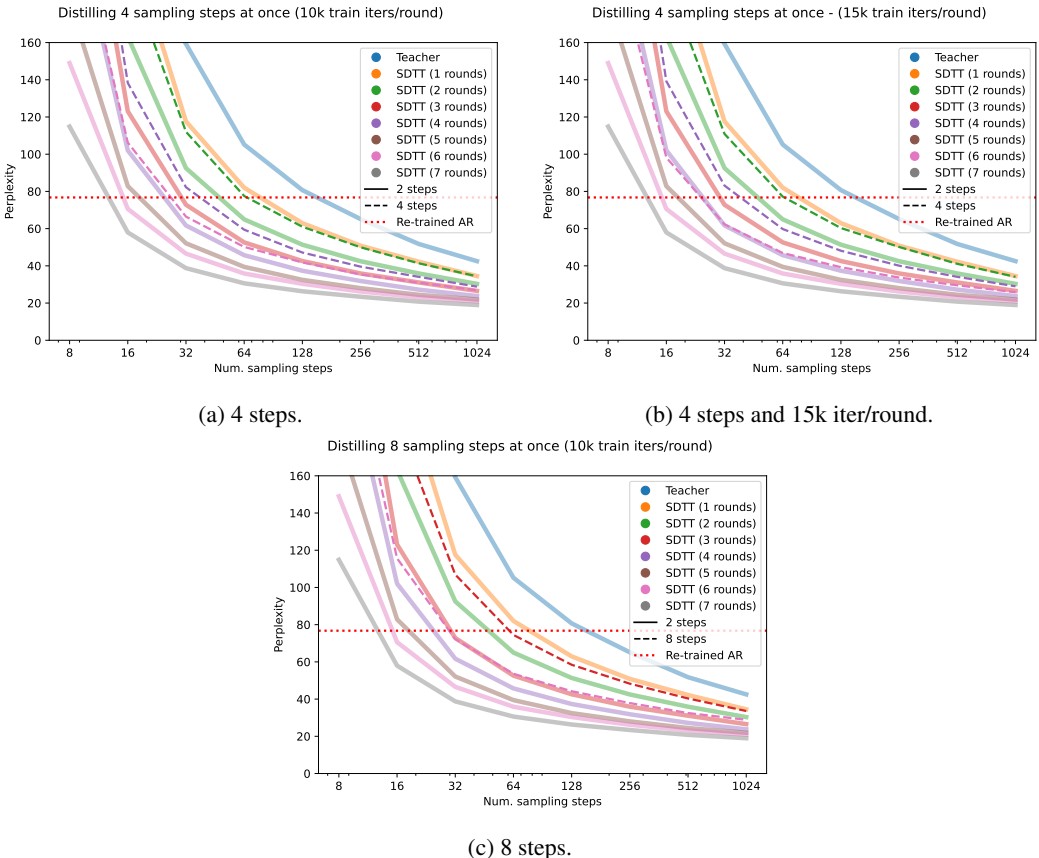

(a) 4 steps.

(b) 4 steps and 15k iter/round.

(c) 8 steps.

Figure 13: Trying to distill more than 2 teacher steps at once. **(a)**: Distilling 4 steps at once. **(b)**: Distilling 4 teacher sampling steps at once wit more training iterations per round (15k). **(c)**: Distilling 8 sampling steps per iteration. Overall, distilling more than 2 steps at a time seem to hurt performance. One could expect that distilling more steps at once would require longer rounds to train, hence we tried growing the round to 15k steps per round, which hurt the performance of the student.

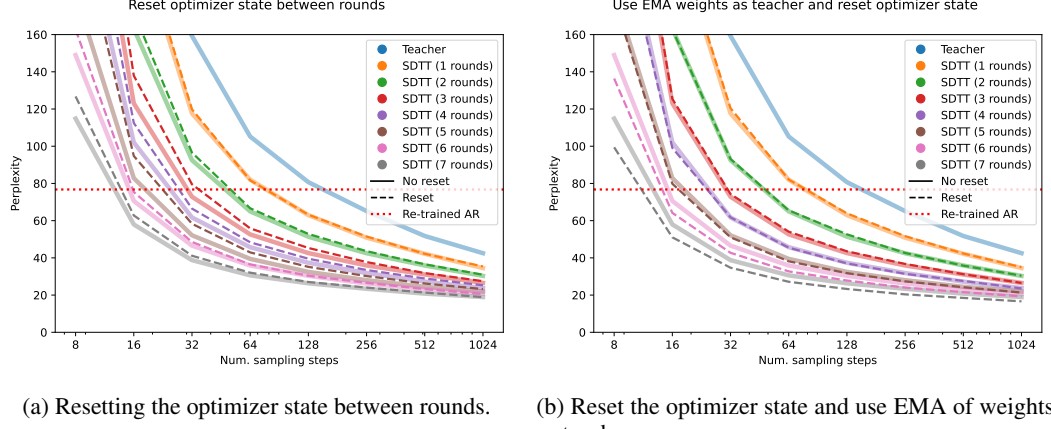

(a) Resetting the optimizer state between rounds.

(b) Reset the optimizer state and use EMA of weights as teacher.

Figure 14: Generative perplexity when resetting optimizer or EMA state between rounds of SDTT.

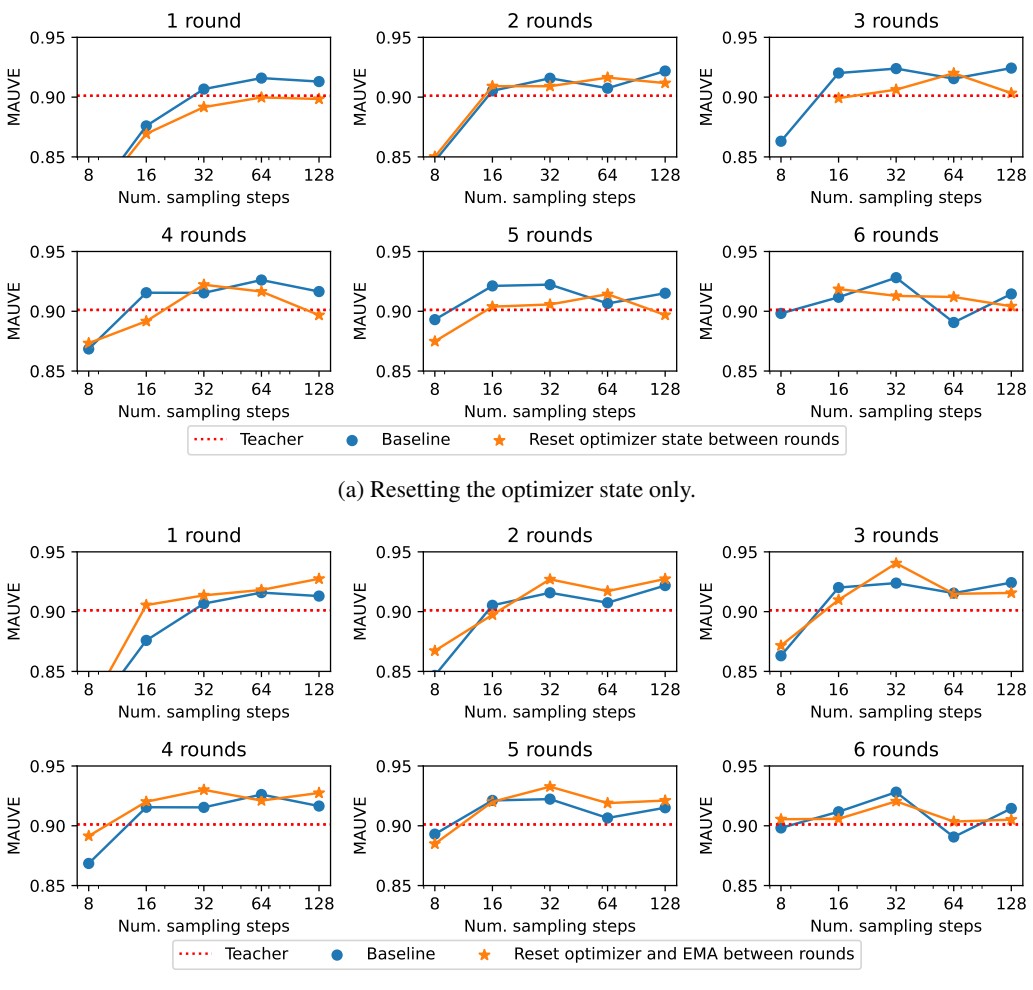

(a) Resetting the optimizer state only.

(b) Reset the optimizer state and use EMA of weights as teacher.

Figure 15: MAUVE performance when resetting optimizer or EMA state between rounds of SDTT.

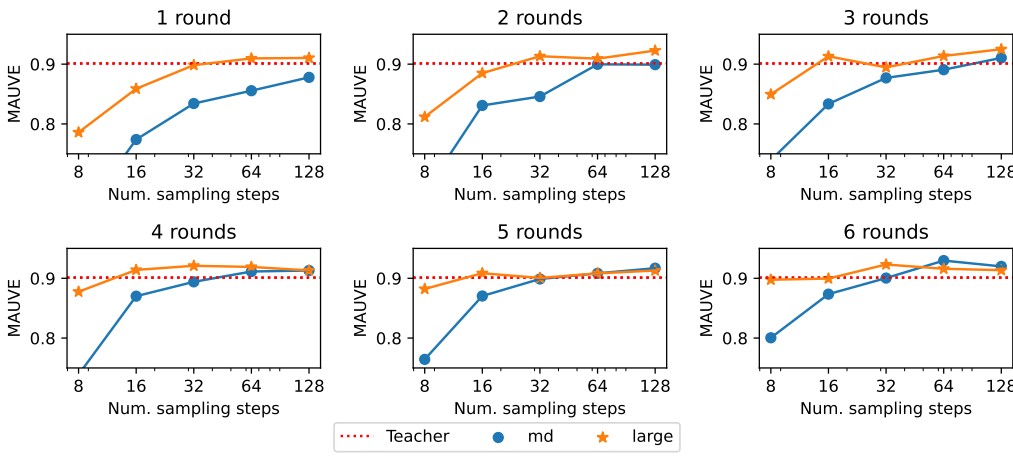

Figure 16: MAUVE performance of medium and large models pretrained for 400k steps. This experiment supports our claims that SDTT helps the final models to approach the performance of the teacher with less sampling steps.

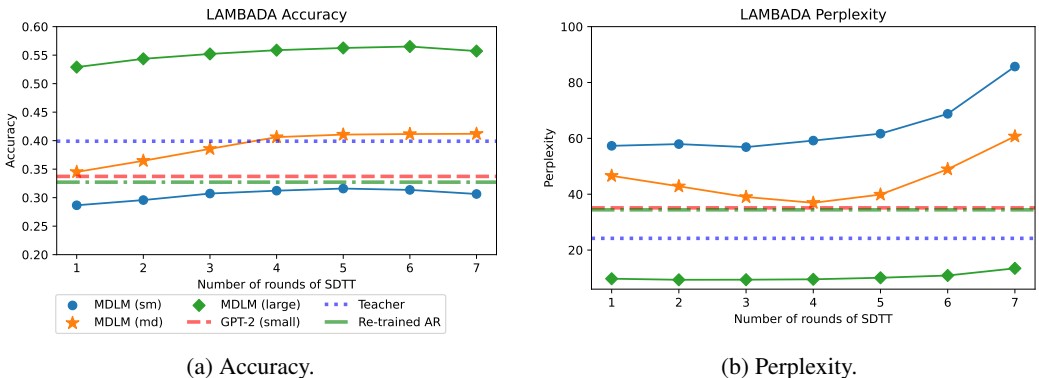

(a) Accuracy.

(b) Perplexity.

Figure 17: Accuracy and perplexity on LAMBADA when scaling SDTT to larger models. All models are trained for 400k steps before distillation. On the small scale, training for 400k steps instead of 1M yields a weaker model. Interestingly, the perplexity can improve after distillation when the models are undertrained.

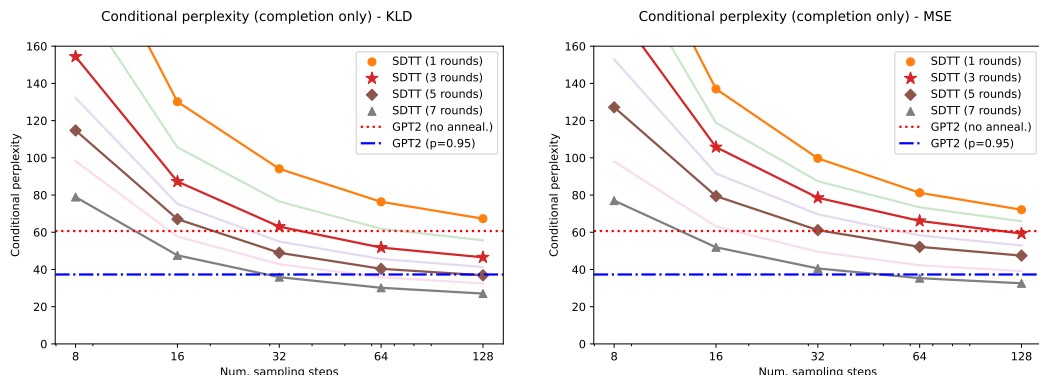

(a) Perplexity of completions when distilling with the **KLD** objective.

(b) Perplexity of completions when distilling with the **MSE** objective.

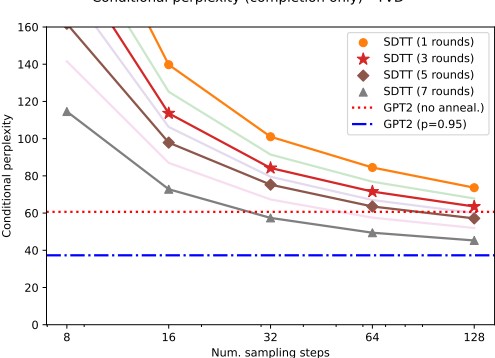

(c) Perplexity of completions when distilling with the **TVD** objective.

Figure 18: **Conditional perplexity.** Perplexity of the completions using GPT-2 large, excluding the prompt. SDTT with TVD performs worse. The final student distilled with KLD matches GPT-2 with nucleus sampling. Ground-truth continuations have a perplexity $\approx 13.11$.

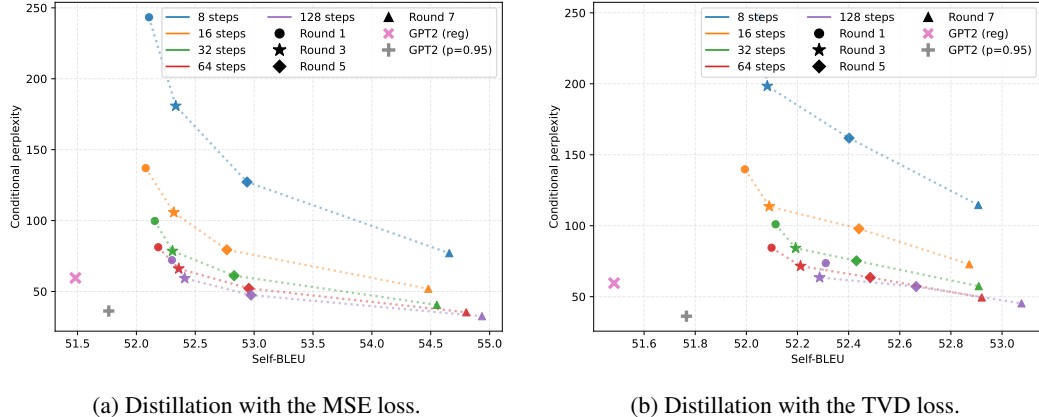

(a) Distillation with the MSE loss.

(b) Distillation with the TVD loss.

Figure 19: **Diversity of conditional generation (small scale).** We measure the trade-off between quality and diversity using Self-BLEU (Zhu et al., 2018). Deterministic sampling yields a score of 1. The diversity minimally decreases after distillation.

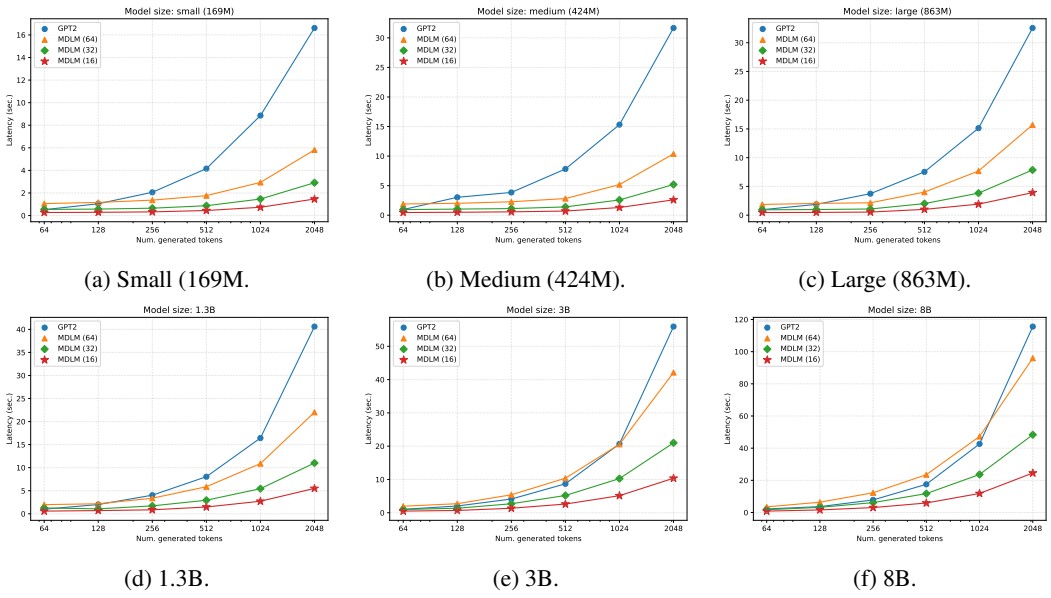

(a) Small (169M.

(b) Medium (424M).

(c) Large (863M).

(d) 1.3B.

(e) 3B.

(f) 8B.

Figure 20: Additional latency experiments with a batch size of 8.

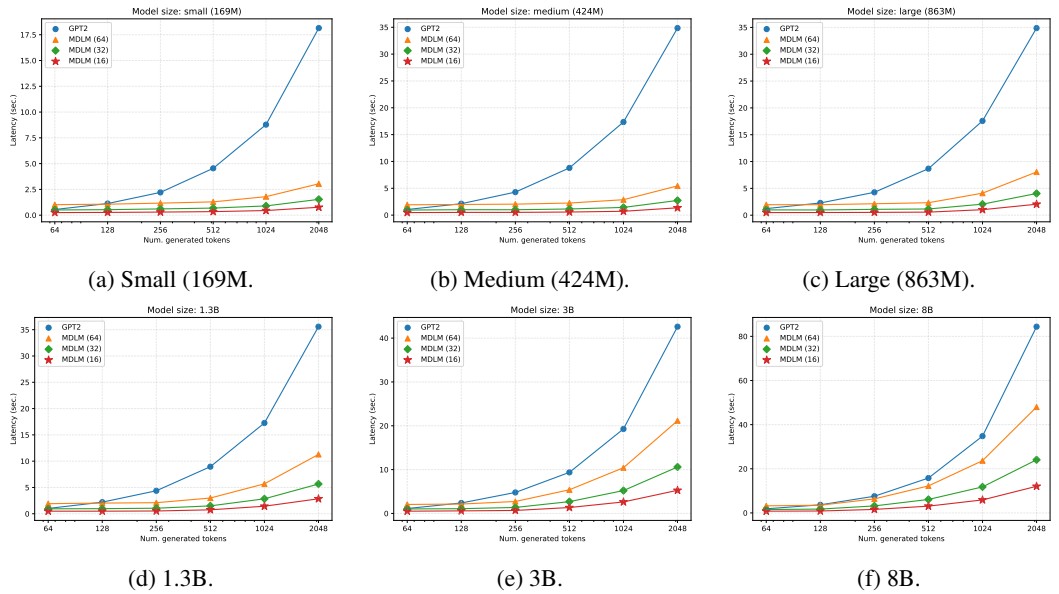

(a) Small (169M.  (b) Medium (424M).  (c) Large (863M).

(d) 1.3B.  (e) 3B.  (f) 8B.

Figure 21: Additional latency experiments with a batch size of 4.

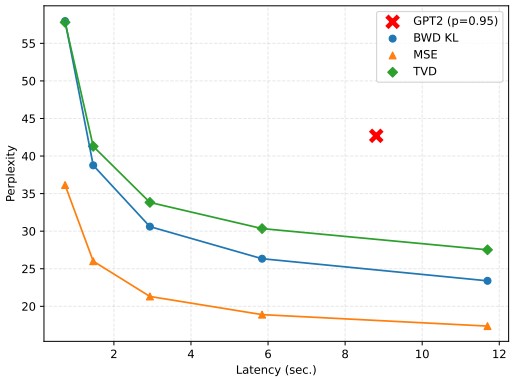

Figure 22: Perplexity vs wall-time latency (in seconds) for small models. We use 16, 32, 64, 128 ans 256 decoding step for the diffusion models.

| Model size | small | medium | large | 1.3B | 3B | 8B |
|---|---|---|---|---|---|---|
| # params | 169M | 424M | 863M | 1.3B | 3B | 8B |
| Num Layers | 12 | 24 | 24 | 24 | 26 | 40 |
| Embedding dim. | 768 | 1024 | 1536 | 2048 | 3072 | 4096 |
| Num. heads | 12 | 16 | 16 | 32 | 32 | 32 |

Table 2: Hyperparameters of the diffusion models at different scales. All models use RoPE positional encoding (Su et al., 2023).

## B  ADDITIONAL DETAILS ON THE DIVERGENCE MEASURES

In this work, we teach the student to match the teacher targets $\tilde{\mathbf{x}}_\theta^{\text{teacher}}(\mathbf{z}_t, t, m/k)$ generated by algorithm 1. We penalize the student deviating from the targets using one of three divergence measure: the Kullback-Leibler Divergence (KLD), the Total Variation Distance (TVD), and the Mean-Squared Error (MSE). We now describe each of them.

### B.1  KULLBACK-LEIBLER DIVERGENCE

The *Kullback-Leibler Divergence* (KLD) between two discrete distributions $p$ and $q$ defined on the same finite sample space $\Omega$ is computed as

$$D_{\text{KL}}(p||q) := \sum_{x \in \Omega} p(x) \log \frac{p(x)}{q(x)}. \tag{9}$$

The KLD has a unique minimum when $p$ and $q$ are equal, however the KLD is not symmetric, meaning that $D_{\text{KL}}(p||q) \neq D_{\text{KL}}(q||p)$ in general. In this work, we train the student with the reverse KLD $D_{KL}(p_\theta||p_{\text{teacher}})$. In the next paragraphs, we present differences between $D_{KL}(p_{\text{teacher}}||p_\theta)$ (forward KLD) and $D_{KL}(p_\theta||p_{\text{teacher}})$ (reverse KLD).

**The Forward KLD**  The forward KLD is called zero-avoiding because if $p_{\text{target}}(x)$ is non-zero but $p_\theta(x)$ is close to zero, then $p_{\text{target}}(x)\frac{p_{\text{target}}(x)}{p_\theta(x)}$ will be large. To minimize the forward KLD, $p_\theta$ will try to assign non-zero probability to all points where $p_{\text{target}}$ is non-zero.

**The Reverse KLD**  The reverse KLD is called zero-forcing because if $p_{\text{target}}(x)$ is close to zero but $p_\theta(x)$ is not, $p_\theta(x)\frac{p_\theta(x)}{p_{\text{target}}}$ will be large. To minimize the reverse KLD, $p_\theta$ will try to assign zero probability to points where $p_{\text{target}}$ is close to zero.

### B.2  TOTAL VARIATION DISTANCE

The *total variation distance* (TVD) is a metric used to compare two probability distributions. For two discrete probability distributions $p$ and $q$ defined on the same finite sample space $\Omega$, the TVD is computed as:

$$d_{\text{TV}}(p, q) = \frac{1}{2} \sum_{x \in \Omega} |p(x) - q(x)|. \tag{10}$$

The factor of $1/2$ ensures that the TVD ranges between 0 and 1, where $d_{\text{TV}}(p, q) = 0$ if and only if $p = q$.

### B.3  MEAN-SQUARED ERROR

Unlike the Kullback-Leibler divergence (KLD) and Total Variation Distance (TVD), the MSE can be used to compare any scalar quantities, not just probability distributions. For numerical stability, we compute the MSE in log space:

$$\text{MSE}(p, q) = \frac{1}{|\Omega|} \sum_{x \in \Omega} (\log p(x) - \log q(x))^2. \tag{11}$$

## B.4 $\chi^2$ DIVERGENCE

The $\chi^2$ divergence can be used to compare two probability distributions. For two discrete probability distributions $p$ and $q$ defined on the same sample space $\Omega$m the $\chi^2$ divergence is computed as:

$$d_{\chi^2}(p, q) = \sum_{x \in \Omega} q(x) \left( \frac{p(x)}{q(x)} - 1 \right)^2 = \sum_{x \in \Omega} \frac{1}{q(x)} \left( p(x) - q(x) \right)^2. \tag{12}$$

As such, we see that the $\chi^2$ divergence is related to the MSE. Note that when using the MSE for distillation, we penalize the error in log space, while the $\chi^2$ penalizes error in probability space. Additionally, the MSE uses a uniform weight factor $\frac{1}{|\Omega|}$ for each term of the sum, while the $\chi^2$ divergence uses a weight of $\frac{1}{q(x)}$.

## C IMPLEMENTATION DETAILS

**Architecture**  To compare with Sahoo et al. (2024), we trained the diffusion models using their code and pre-processing steps on the OpenWebText dataset (Gokaslan & Cohen, 2019). As Sahoo et al. (2024), our models are not conditioned on the noise level. Nonetheless, Sahoo et al. (2024) kept the architecture of Lou et al. (2023) unchanged and makes the model unconditional by feeding it a zero tensor instead of the noise level. Removing the adaptive layers could improve the sampling speed further, but we avoided modifying the architecture to prevent potential problems. See table 2 for the hyperparameters of our models.

## D TEXT EXAMPLES

We include non-cherry picked text generated from the small distilled model with KLD loss from the last round of distillation via unconditional sampling with varying number of steps. We show the first 512 tokens to so that the text fits on one page. Remember that those models are small and not fine-tuned for text quality. They can also start generating in the middle of sentences, since they are trained on a concatenated corpus of documents.

Text generated with 16 steps (1/3)

```
invite to the gathering, because he was invited in 2008,
    probably on a regular basis thereafter. But to become a
    scientist, to verify those cred veracity is important,"
    inlamali said.

CNN is thus creating a monster that has supporting cascade of
    other grand jury investigations, he said.

"In the case of Mr. Eliaschis, I wrote in a today to a number
    of everyone involved in consideration of this matter;
    these people are invited, named and considered 'committed
    ' to the process and trust of the Nation," he said.

There have been no complaints or formal complaints and this
    will directly no longer be CNN's standard and indepth
    coverage.<|endoftext|>because my office cherish diversity
    , this is the approach we have come across. their
    subscription model is great, and right now there are
    folks in our office that want to help to promote
    diversity. so we're looking forward to hearing those
    responses from them. although we realize it is a
    different place than we run it. but outside of this, I
    think we've never had a lot of conversations (especially
    here) about community-based leadership being happier than
     market-based leadership: the leader is fantastic, the
    person is valuable and talented.in the UK things don't
    that way. it has a leaderless culture which has not been
    well with a hierarchical planning and organizing process.
     we have a very specific image of this kind of
    organization. but one of such qualities is the image of
    someone accomplished like everybody else does, which
    interests us as do the talks of one of our public figures
    .

as part of what we'd love to do here on our social impact
    endeavors. recognize that most of the work here, we're in
     the midst of the first day of the interview (which Prof.
     Garry had posted to the blog). Garry was kind enough to
    come participate in the interview as well as conducting
    and perusing on his and the next few competitors, in
    order to get valuable feedback. I wanted to feel
    enthusiastic about the process, eager to share feedback,
    and expect to have a very professional experience.

but, broadly speaking, more than a lot of the things that we
    struggle with the capacity to report, we just made a
    draft, and then got the post called. there is one example
     of things throughout the draft that made me the most--
. The media is a small part in Far Left.

it is really important to have a relationship with your
    employer. At a high
```

Text generated with 16 steps (2/3)

```
kids not being in our schools as a result. The kids put their
    families in Florida schools in this district, this is
    Georgia school, and not only do they have enough time to
    work for a Florida firm, it's not desirable for our kids
    to be in Florida school anymore."

The brothers turned to public education and the governors
    quickly asked them if they wanted to. Bonding Aid then
    was contacted once asked for a special order from the
    administration setting aside $524,000, but they were
    denied despite the requestBy Bill Othello, according to
    government spokesman. The brothers wrote letters to the
    governors numerous times claiming the information was
    false, including one letter, which suggested that they
    funnel $2,000 to the Slothouse Clearing House schools
    through the University of Florida. However, one of the
    brothers, Chris Yates, told Bloomberg News that he still
    was shocked and horrified by the correspondence, saying,
    "Obviously, I felt like a coward to be in this of a very
    uncomfortable situation."

Instead however, Yates said, he reacted very much like he
    pissed off at one of his favorite politicians, Bush.

Florida's two Republican leaders have strained relations, in
    particular with recent governors Doug Ducey, a Republican
    , and Jeb Bush, a Republican, addressing his concerns. "I
     do not think the other leaders will do this, but we do
    have to work to make sure that is how we have to do it,
    something that's something that we need to be doing, and
    that there will be always a need for better quality
    education, too," Bush also said. "We do not want Florida
    to stop funding education and essentially contradict the
    fact that what has a provenance in this planet is that
    killing was at least 10 percentage and many people got
    dead. We I beginning to have problem with that and I
    think that's the first where we know, for sure, when we'
    re going to eventually share that information with the
    public, and we feel in order to deal with that we have to
     agree to the efforts that we are engaged in and also [
    Haley and I] are ultimately going to have to provide that
    . We will have the heavy lifting to do whatever we do. I
    think this is irresponsible but also that we are not
    going to start having a real conversation because we've
    got a lot to do, but that was a eye opener to us."

When asked on Tuesday if he was dazed and that he regretted
    attending the meeting
```

Text generated with 16 steps (3/3)

was subjected to will then acidize in the form of foam. The pH and pitch of foam create a, too substantial a gap between an egg and tissue, which will drive it to accumulate faster, and can thus cause irritation to the sensitive parts of the body. It's penalties are well as analgesia, and shortens chances of learning how the material works.

Mr. Segal, who was involved in the study, did not acknowledge the limitations of the study to treat his own specifically painful dental fracture, but added that it succeeds in all aspects of the process. "This is the first time that it might make a significant contribution to improving dental health. Until then we will have to get better at adjusting what have changed to make sure that it is effective."<|endoftext|>Yet, in June 2013, Laquan Phillips, 20, a promising medical student at the Jackson State University School of Medicine and the son of a cyclist Philando Castile, killed two or four weeks earlier, was forced to give a the dozens of officers. officer just about three seconds to pull up on the vehicle trying to lock the black car into silos, and Fewell, the neighborhood managed by the officer who arrested Phillips, was refusing to go all the way. The officers were concerned about the direction of bullets so that they could hit a casting bit.

While police believed the bullet hit a man on the left side – and a belief that had been consistent since police had been in deep denial, it failed to hit the man on the right side of the table. When you interjected a shot into the man's first body, the face mostly rolled down the throat – a crushing moment of motherhood and last sweat for a father of two – and thankfully the other one was stopped in his tracks.

"We believe him," said John Milliken, a police officer at the time of his shooting, who could not confirm other deaths but said that Oli was firing gun. Others would say it was the product of head trauma.

The policeants in Roswell could rely on a variety of lethal weapons, they said, including ammunition that police had accessed. I want to thank the people for the first– aid kit for the family, and the people and the people for Justice Jesse James also, at hand members of the Black community the 71st.

An anonymous person was having a phone conversation with the Buckeyes's interim president, who was to take part in the participants of

**Text generated with 32 steps (1/3)**

Wilkins was he committed the acts of vandalism as a juvenile
.

Woodward said he had "since confessed the crime based on
information and an explanation for what he did." But a
police spokesman told the news outlet that investigators
work for the government and it is the responsibility of
whether it is the individual with knowledge of the crime
or that should be punished as well as those involved with
the system.

"Liberals can withhold confidential information from the
public on the basis of any reason or whether such
information is a public interest," the spokesman said on
condition of the anonymity because of the investigation.

Fox News reporter Brian Stelter, reporting the agency's look
at the alleged Hammond scheme, took a high-profile line
on political campaigning in Russia and current affairs on
Fox News Channel's propaganda channel, The Which?," for
instance.

The Rasmussen Reports cited concerns of a spike in voter
fraud, citing the divide between the Democrats and voters
, many of whom Republicans outside the party vowed to
lose in the election.

The Washington-based advocacy group We Are America, which
publishes figures from the polls, said the report was
based on the promises of conservative news outlets,
independent organization, and the use of pollsoddy
reporting.

"The communication and reporting of American media has been
critical," We Are Russia said in a statement following
the report.

The United States media has been biased against the election,
which Russian officials linked to hacking on Democratic
computers and other promotional efforts that seem aimed
at raising security fears in Moscow and toppling many of
the Kremlin's market allies in the U.S, which is moving
closer to winning the election before a referendum begin
taking place.

Russian officials have attributed their bias in opinion polls
toward the 2016 elections to an uptick in unemployment
statistics, with figures they're publishing being
forecast on the eve of their presidential elections.

Russian officials say they would like to prevent fraud, and
polls show that they may have aggressive on potentially
maintaining a narrow five-point lead.

The Rasmussen poll report came days after the state Attorney
General filed a lawsuit in general against the Rasmussen
Reports' report, and asked the state to look to the
opinion polls and determine if the Rasmussen poll did a "
good job."

Democrats announced plans to use a federal court case to

Text generated with 32 steps (2/3)

Sherman was a city historian for 28 years before 1988 and
    former relations adviser spoke with more than a dozen
    Asheville staff members during an interview outside the
    hotel early Monday morning. In a brief interview, Sherman
     gave employees an overview of what they seerved from the
     two Highlanders – including a handwritten piece of
    electric paper and pictures of the cut and logings,
    perhaps sweeter than the compensation Yancey and Davis
    did compensation settlement for.

Photo: Courtesy U.S. BAG via Jan. 30; The documents provided
    in the report stem from a link to a minor change in the
    law, according to the report.

The law officially establishes an enforcement and oversight
    process (PDF) for transferring payments to the other
    committee, or the City or State Board, that has the
    mandate to require the cash transfers from the other side
    ;

The legislation, upon meeting each committee, instead directs
     the legislature to set its own financial policy in the
    other committee;

The method suggests payments to the other committee can form
    the legislature's own committee to pass enforcement and
    oversight legislation, and that each of the committees
    may vote on motions for a resolution and give the public
    a vote or motion to approve legislation by the Joint
    Finance Committee.

That, the city and state will follow the same process as the
    New Hampshire City and Lodging Act. However, in the Nov.
    election the Legislature prepares to decide whether to
    modify the rules and procedures surrounding such a Senate
     bill, so it would be unclear how long the interim
    Legislature policy to be implemented once it's adoption
    is in place.

The terms of the report are not necessarily overarching, some
     of the other ways are specifically saying that a
    proposed legislation that would transfer payments to the
    other committee must be amended to include a general bill
    , before it would be recognized as a legislative body.

in Khan at editors@time.com

Twitter: @in_khan<|endoftext|> independent party New Greens
    has urged Parliament to declare that there was no reason
    for the war. Many Australians who believe that it would
    be better to their children going to war are handed local
     police officers the chance only for training next week.
    The leader of South's Republic of the African state,
    Christopher Robertson, said that the government's refusal
     to give any status to the force is the worst for
    Australia's history. In a statement to the Liberal party,
     Mr Robertson said it is important to keep a police force
     on ground as it sees fit. This has caused panic among
    all 16 states, in particular the growing

---

**Text generated with 32 steps (3/3)**

```
, the police, media and public

International Council of the Community of Europe (IECE) is
    preparing the next steps, being taken on Monday – that
    could the potential to make firm changes to Britain and
    the EU, including the introduction of modern regulations
    and procedures on trans financial transactions from
    ordinary individuals to private corporations.

However, senior mainstream EU members told the Telegraph that
     some of the changes in the legislation at the moment "
    are not clearly relevant to that role" for regulatory
    oversight, as well as legal procedures, for the provision
     of business and financial industry, social, welfare,
    police, promotions, public health and economic activities
     – and finally, to give the EU the right to make the
    legal and regulatory decision that is needed by the wider
     market.

A senior Council official in the UK, speaking on condition of
     anonymity for the sensitivity of the meeting with his
    staff, said there had been little progress in negotiation
     negotiations and how to implement the changes because
    the legislation was still on the way. The senior official
     said both countries had worked fully with each other so
    the legislation could meet the basic existing EU
    regulations.

"We agree then that we will need to amend the legislation in
    a way that is appropriate for EU law in the parts of the
    UK. So review and consultation is something we are
    looking very closely to to ensure that our new amendments
     to the legislation give additional leadership and
    framework ... to verify we can implement those measures
    and will provide appropriate support of expertise to the
    UK in relation to getting them."

However, the official said it could take some time still to
    implement the legislation and that any changes put in
    place require a complete clean review of the wording of
    legislation, and part with EU institutions in that way.

"We understand the process that is underway on how to
    implement those measures – and implementing those
    measures is going rapidly, so we can't yet start to
    assess the situation," he said.

The Council has agreed on January 25 to pressure the UK
    government on the decision to leave the EU to good effect
    .

"The previous government had helped reform the law around the
     UK and contributed to that change," said Matthew Kennedy
    , an official for the Council.

The Council will work on the new amendments as part of two of
     the terms of engagement with the British government,
    signed by David Cameron.

"They are part of a broader effort and to do so has not been
    determined, so the Prime Minister and those
```

Text generated with 64 steps (1/3)

```
create big gas storage wells locally. BP has proposed
    investing billions of dollars to build new big storage
    wells, which will raise gas production to new levels of
    production.

The research has been co-sponsored by Congress lobbying to
    oppose oil and gas drilling (FAPL), but the
    administration has yet to outline how far oil and gas
    drilling and the Eagle Ford expansion will take its new
    forms, with new risks rising.

SUS energy industry officials say changes in the measures
    ordered and new funding levels have been little bearing
    on oil production, capped in about 60% last year, but
    some proposals are still receiving orders - still others
    have yet to be overcome - by the Department of Energy to
    produce a package on a proposed Dakota Access oil
    pipeline to the West in the West Coast.

The distinction between fossil fuel and gas is also blurred
    on energy measures that have been laid out to Congress
    about a year; most are now in the process to agree on the
     proposed Keystone XL pipeline, and the Dakota Access
    pipeline.

The US energy market is fast moving, so the focus of the
    Obama administration has been to push jobs to the US and
    to encourage Americans to have better financial prospects
     for creating their new jobs

Gans to set low-carbon renewable energy targets will soon
    launch at the White House. Since 2009, infrastructure
    projects have established several significant initiatives
     : restructuring the coal sector in the West of Europe,
    reducing the tariffs on renewable energy in Britain, and
    implementing zero-carbon policies in South Africa,
    despite high difficulties in succeeding in implementing
    EU emissions standards.

The infrastructure initiatives, announced and unveiled by the
     president on Wednesday, are focused on the aim of
    finding effective ways to build on competitiveness at
    long-term scales and to help around the world create a
    sustainable energy portfolio in the areas needed to
    mitigate the risks in local economies.

But institutions as big and large as central banks
    infrastructure plans to support renewables have been
    especially critical in recent years. A number of EU
    states such as Finland and Sweden were taking initiative
    with their plans. The states, which will also launch a
    number of the other initiatives at the White House
    ceremony, said that "the energy policy environment will
    receive high profile development in the UK and the
    economy of the EU in most of them".

The far-reaching strategy has been to place low-carbon
    targets in the UK's renewable energy sector, of course,
    with a push of the private sector - much of which is
    being led by governments and businesses - to scale their
    renewable energy
```

Text generated with 64 steps (2/3)

exchange Bitcoin, it should be able to deliver such services. Basically, a service that hinges on Coinbase make payments for those bitcoin users who use this unique means of purchasing and storing bitcoin, and that it's able to reach retailers as wide as possible, which makes it so attractive for merchants for transaction speeds. Once they've taken the step of overcoming a long wait for an WePay service, they're permitting merchants to accept Coinbase to begin. What explains the delay is that even if they decide moving forward with an offer to pay for maintaining the price of bitcoin and the impressive growth of bitcoin, it remains a high barrier and additional cost for some users. What's sure for certain is that merchants will accept the service, but that it will deliver a version of bitcoin as a method of payment.

They also plan to allow Coinbase to maintain control of the new address, to ensure the security of the integrity of Coinbase, as well to further expand the service available to Mac users. And besides, Coinbase is still investing in BitPay, a leading bitcoin exchange offering for free an alternative to fiat value, which would be a perfect fit for a marketplace for all that bitcoin in a heavily regulated world.

We have reached to WePay Bitcoin for our response and for their views on our WePay service.<|endoftext|>How do you describe the typical user of marijuana. Your questions are always open to the cipscoop.net email team. Today is Day of Cannabis, an event celebrating the global outside of the marijuana industry. Let us know your favorite questions.

K: Let's talk about the for-profit group that supports marijuana, Gives of Health, which established the first nationwide standard price system for marijuana. Can you name a name?

In its first statement, Gives of Health defined itself as committed to supporting a national free market system of marijuana, promoting policies to reduce the supply of legal prescription drugs, reduce tobacco prices, reduce wasteful government spending, and taxing and regulating marijuana. We believe a marijuana marketbased tax system is good for our public health and for the economy by helping people circumvent drug laws and purchase marijuana, reducing tax costs associated with prescription analgesics and avoiding tax evasion.

K: Leon and several advocacy groups that try to crack down on marijuana, as the Gang of Eight, introduced bills to make cannabis and medical marijuana for adults in more than a dozen states. What is that to say about addressing a lot of the problems?

Text generated with 64 steps (3/3)

```
by the military. The army (which elected its President) saw
    the most important coup in its history.

The broader American Catholic Church claim that one of the
    first indications of the twentieth century thought of "
    liberated Christians" not earlier than July, 1960 was
    that the Protestant church would be leveled within their
    territory. Protestant mainline hierarchies in the late
    1950, and early 1960, which had witnessed the life of a
    Calvinist Protestant movement, built the Protestant image
     of a new creed that preached tolerance and loyalty to
    the Church, desperately looking for a new denomination.
    Although the political movements had been building up for
     a long time, there was a Protestant political movement
    that thwarted the church and introduced a sense of
    impending doom specifically to a certain movement, a
    belief that while the Reagan government's attack on the
    Church made it Americans personally and at the center of
    society, and eventually dissolved and destroyed
    Protestant institutions. The church, with the exception
    of the United Methodist Episcopal Church (MCCA), heavily
    engaged in Latin American, Australian, and indigenous
    communities, is recognized as having one of the first
    Protestant hegemonies in the United States, following the
     formation of St. Mary Catholic Church in Louisville in
    1963.

The circumstances of the resurgence of radicalism have led to
     a political movement that introduced a distinctly
    Protestant identity and spread feelings of insecurity and
     alienation that today plague converts to Christianity.
    This is unusually widespread within the broader American
    Catholic Church. Often associated with radical movements,
     such trends may signal a significant shift in the
    Catholic Church from the separation techniques throughout
     the history of the United States Church. The Southern
    Baptist Convention Conference, established its Center for
     the Restoration of Christ of God, in America. Later, it
    established an Adventist church in Texas. The Maclean
    Church professes and operates another Adventist church,
    the Church of America, which began forming one of the
    most centrist and largest denominations in the country in
     1960. As a result of this, the largest denomination
    would soon be Holy Cross Church in North Carolina. Nick
    Sheymia, president of of Faith in America, which
    maintains a hard line between conservative and
    fundamentalist Christianity on the left, told the New
    Apostolic Arrangement Network that the church has always
    "walked in the overall shadow of the much broader church
    that attempted its rule and control of the nation."

Ol Howard, an educator and the author of the American
    Catholic Church: A Study in Truth, said, "The church
    seems inept at the end of the time. But to most Americans
     it
```

Text generated with 128 steps (1/3)

```
most countries, but in the UK, where computers are run by the
    as many as 15 people involved is more complicated. Many
    experts think that governments may need to overhaul their
     machines because they are unable to retain their
    influence because budgets are stretched too thin. In such
     a complicated situation, they say, public sector efforts
     to move ahead in technology, particularly climate change
    , would have to be put in real action.

Another major concern was a government effort to undermine
    the private sector in the construction sector and
    financial services industry. Ben Gove, chief executive of
     the House of Commons, said the government needed to
    address the role of the public service sector, and made
    explicit that the role of the labour force has become an
    effective vehicle to cut income from the poorest working
    people to the richest.

20%: the economic consequences of mass industrialisation Read
    more

In a joint statement with several coalition partners, the
    Conservative party also called for changes in the public
    sector to ensure the return on public investment, as well
     as the budget. Labour pushed back against some reforms,
    which involve modernising the benefits system for older
    people to make it easier for businesses to compete to win
     jobs. However, these reforms critics say apply only to
    the people who need them.

Ben Smith, the shadow environment minister with a record of
    climate change on the Labour side, expressed "general
    disappointment that this report shows significant
    shortcomings". In practice, the Commons report is not
    meant to advocate for such reforms. Labour would have
    hoped to represent a party with a strong alliance with
    existing policies and a government that has committed to
    investing in more workers overall.

Labour responded by saying: "We're pleased the MPs voted to
    put in place further policies to help the workforce and
    benefit everyone."

Labour paid for the narrow, decisive vote with a single vote.
     They say the findings show the government has missed the
     importance of its work for helping the working class
    within the economy, so that the enormous impact of
    automation benefits is linked with the exploitation and
    marginalisation of the working class.

The government has failed to recognise the fact that energy
    profits have been rising. Since 2008, bosses have tripled
     the ranks of the British electricity and gas companies,
    helping their annual profits rise by 3.1bn over the
    previous decade.

The biggest companies have made more profits over the decade
    before. This has created an incentive to make for more
    secure and sustainable energy security. But as well as
    bosses, they must collectively recognise what is
    happening,
```

---

**Text generated with 128 steps (2/3)**

```
many really exciting, non- independent artists, and all these
    indie bands, jazz and folk-rock, and rock-and-roll jazz
    bands, and you see a lot of new artists making these
    kinds of fantastic music. What do you think about these
    artists in Seattle?

M: I my whole life have never really wanted to do anything
    outside indie music, so I wanted to go way mainstream. It
    's much of a role model to try and spread the word. I
    think everyone gets invested in it so that people can use
     their music, but even back then, nobody had tried to
    bring these kinds of music to that community, and I saw
    that it was a good way you should find someone who's
    going to do that in the right way. This city can be an
    extremely helpful tool for developers to come up with the
     right media resources to develop quality music, and now
    there are so many indie studios over the country just
    attracting artists, and I think it's very helpful, and so
     there are a lot more open tools available.

S. Norman, what is probably the biggest impediment you put in
     terms of the music scene, and what's really important
    for you and I to bring your music to audiences in
    Portland or New York?

M: I would say the only really obstacle to me, it's the fact
    that I have to work with people in Seattle, and I try to
    involve them in it as much as possible because of where
    they actually come from. Many of the people we were
    talking to are basically part timers in the industry, and
     that I and I have had to work with, so many people are
    also a part of the community, among all of the audiences
    we've been meeting. And I really came out of this scene
    with a lot of excitement about it. And because I grew up
    growing up here in Seattle that was crowded and full of
    loud noise, it had just turned out to this place with so
    many people in this scene, that I found myself moving
    more to the music that I come from, at least to me, in a
    small, vibrant, and growing community, and it makes that
    sound more accessible to everyone. And what is really so
    encouraging is that while most of the people that we were
     talking to at events, they are also taking part in
    actions within the community, supporting our music
    through connections and sharing it with each other in our
     community and the city. Right now, there's really not a
```

```
Text generated with 128 steps (3/3)
```

```
to write this because it blew me out of my seat. I've been
    getting a few negative reviews in the past couple of days
     and I realize that people are much harder than ever to
    read them. Well, everything I've heard to this is
    overwhelming, and I'm happy to say that there am so much
    that I know that has really blown me out of my seat that
    this situation is much worse than anything I have told
    you. I want to be very direct to you, regarding even some
     of the things that you have heard. Feel free to let me
    know what you feel like — whether I have a personal
    concern or a sincere thing you would like to hear about
    this topic.

The response to this story is overwhelming support and love!
    I just want to reach out to my Facebook page to tell you
    what I I wish to do for you and really appreciate. I don'
    t want to talk to anyone just to tell you how I'm very
    sorry. I want to tell you about all the things that I've
    been through so far and what I've done for you. I'm
    really sorry for everything that I have done to this
    community, and I am really appreciative for my support.
    Not to even go into detail the issue too much, please do
    head over to the next page in the post.

I ultimately don't want to tell you that this is your biggest
     issue — I don't assume that it help you much. Instead, I
     want to say that you should just get some things out of
    the way. I've encountered many other people just coming
    forward to this, and the fear of losing anything that I
    try to say about these issues can only interfere with
    such a process. However, now I like being out there
    trying to help the world, and I am so thankful that the
    community might help me too. Although I try to read this
    as about once a day, I just appreciate the amount of your
     feedback and hopefully I can build more positive ones to
     read by the evening.

Lastly, I want to inform everyone regarding this topic in
    case that other readers want to buy my blog here....So
    hopefully when I write something to my readers in order
    to help them out, I like your support. I want to support
    every reader so that I can talk to other readers for any
    information that can help me, to further tell my story.

C: I still have the horrible feeling that I
```

