# OpenReview forum: "Beyond Autoregression: Fast LLMs via Self-Distillation Through Time"
_ICLR.cc/2025/Conference — ICLR 2025 Poster_

### Official Review · Reviewer_fzmS · 2024-11-02

**Soundness:** 3
**Presentation:** 2
**Contribution:** 3
**Rating:** 6
**Confidence:** 4

**Summary:**

This paper explores progressive distillation, a technique commonly used in continuous image diffusion models, in the context of discrete diffusion language models. Starting with a trained discrete diffusion model that might take many denoising steps to generate a text sequence, the proposed framework iteratively distills the slower teacher into a faster student diffusion model, effectively halving the number of sampling steps in each distillation iteration. The distillation process involves collecting predicted logits for all tokens from the teacher model and then training the student diffusion model to match the teacher’s logit predictions according to a defined divergence measure. This self-distillation yields a significantly faster diffusion model, capable of generating text with up to 32 times fewer sampling steps, all while improving generation quality.

**Strengths:**

- This work makes a valuable contribution by extending progressive distillation techniques to diffusion language models, achieving much faster generation with improved generation quality.
- This work also includes extensive ablation experiments that carefully examine each design component of the distillation framework, demonstrating the solid empirical performance of SDTT. These results offer useful insights for future advancements in discrete diffusion language models.

**Weaknesses:**

- The paper appears to be written hastily and lacks clarity and organization. Beyond typos and formatting issues (examples below), key concepts are introduced ambiguously. For instance, only in the experiments section is it clarified that SDTT consists of multiple distillation rounds rather than a direct distillation; and the progressive nature of SDTT is not immediately evident in the main text or Algorithms 1 and 2, becoming clear only in experimental results. In addition, the number of distillation iterations can only be inferred indirectly from `dt`, which is quite unintuitive and causes more confusion. It would greatly improve understanding by emphasizing the progressive formulation early in the methods section, and annotating the number of distillation rounds explicitly instead of relying on `dt=1/k` .
- Inconsistent use of x-axis labels in figures also adds to the confusion. Some figures, such as Figures 1(a), 1(b), 8, 9(b), 15(a), and 15(b), use “sampling step size” as the x-axis (e.g., `1/512`) while others use the “number of sampling steps” (e.g., 1024). If these two metrics are equivalent to each other (e.g., 512 sampling steps correspond to a sampling step size of 1/512), unifying the x-axis to the “number of sampling steps” across all figures would greatly enhance clarity; and also prevent confusion between terms like “sampling step size = 1/512” and “dt=1/512”, which, despite similar notation, have different meanings.
- In Figure 2(a), the relationship between the figure and various distillation rounds is ambiguous. If I understand correctly, “Round 1” and “Round 2” in the figure actually denote different denoising steps within a single distillation round, rather than separate distillation rounds.
- The concept of progressive self-distillation is not new within diffusion models. The proposed framework SDTT largely resembles progressive distillation [1] with modifications such as the construction of targets and distillation objectives, both of which are straightforward following the formulation of discrete diffusion. While SDTT adapts distillation techniques to a different application and presents strong results, this paper does not sufficiently discuss its related work [1] throughout the introduction and the methods section, making it difficult to evaluate how SDTT differs from previous contributions [1].

[1] Salimans, T., & Ho, J. (2022). Progressive distillation for fast sampling of diffusion models. arXiv preprint arXiv:2202.00512.

**Questions:**

1. Typos:
    1. Line 96: `(MDLM.)` → `(MDLM).`
    2. Line 785: `the the MAUVE` → `the MAUVE`
    3. Line 785 `resuts` → `results`
2. Clarifications on contributions:
    1. In Line 79, “Unlike many distillation methods for continuous diffusion models, SDTT does not rely on deterministic mappings such as DDIM” might require further elaboration. Given that discrete diffusion models inherently lack a clear connection to deterministic processes akin to DDIM, what is the implication of this distinction? Specifically, how does the absence of deterministic mappings influence the theoretical foundations or empirical performance of SDTT?

---

> ### Author Response · Authors · 2024-11-21
> **Authors answer to the Reviewer fzmS**
>
> We genuinely appreciate Reviewer fzmS for their insightful feedback. Below, we present our responses to the questions and comments of reviewer fzmS.
>
> > For instance, only in the experiments section is it clarified that SDTT consists of multiple distillation rounds
> >
> - We understand that introducing the iterated SDTT in the experiments section may cause some confusion. To address this, we have updated the manuscript and now introduce iterated SDTT in the methodology section.
>
> > The progressive nature of SDTT is not immediately evident in the main text or Algorithms 1 and 2
> >
> - We apologize for any confusion regarding the progressive nature of SDTT. To clarify this aspect, we have revised the introduction, updated the title of Algorithm 2, and expanded the methodology section. These changes emphasize the progressive nature of SDTT, which we hope will provide a clearer understanding of our approach.
>
> > It would improve [the submission by] annotating the number of distillation rounds explicitly instead of relying on dt=1/k.
> >
> - We acknowledge that the notation could confuse some readers. Therefore, we have updated all figures and removed all use of the notation `dt=1/k`. We are grateful to Reviewer fzmS for helping us to improve our submission.
>
> > Some figures, such as Figures 1(a), 1(b), 8, 9(b), 15(a), and 15(b), use “sampling step size” as the x-axis (e.g., 1/512) while others use the “number of sampling steps” (e.g., 1024).
> >
> - We have updated all figures following your comments. We hope that the updated text is clearer.
>
> > In Figure 2(a), the relationship between the figure and distillation rounds is ambiguous. If I understand correctly, “Round 1” and “Round 2” in the figure actually denote different denoising steps within a single distillation round, rather than separate distillation rounds.
> >
> - You are correct that in Figure 2(a), "Round 1" and "Round 2" refer to denoising steps used to generate teacher targets for a single training step (i.e., one parameter update). We acknowledge that the use of "rounds" in this context may cause confusion, as it is also used for iterated SDTT. To clarify, we have updated Figure 2(a). We hope you will find the figure clearer (Figure 3a in the updated manuscript)
>
> > The concept of progressive distillation is not new within diffusion models. This paper does not sufficiently discuss its related work [1] throughout the introduction and the methods section.
> >
> - We acknowledge that progressive distillation is a well-known technique in continuous diffusion models. We have updated the introduction and methodology sections to further emphasize the progressive nature of our approach.
>
> > In Line 79, “Unlike many distillation methods for continuous diffusion models, SDTT does not rely on deterministic mappings such as DDIM” might require further elaboration. What is the implication of this distinction?
> >
> - Most continuous distillation methods depend on a deterministic mapping from noise to data, typically using the Ordinary Differential Equation (ODE) form of the reverse process. This means that once the model is trained, it can define a deterministic map (e.g., using DDIM) from noise to images, making it easier to learn this mapping in fewer steps. In contrast, discrete (absorbing) diffusion models do not have such a deterministic map (lines 80 and 462). The noise distribution in these models masks all tokens, meaning only a single sample can be generated deterministically.
> - This distinction is important, as it means we cannot rely on matching a deterministic map in our distillation method. Instead, we must address the challenges of working with stochastic targets. As we hypothesize on lines 363-364, this stochasticity is likely why distilling more than two steps at a time yields worse performance empirically. We have updated the introduction to discuss this further (lines 80 to 83).

---

> > ### Comment · Reviewer_fzmS · 2024-11-24
> >
> > I appreciate the authors' comprehensive responses and efforts to address my concerns. The presentation has now significantly improved and makes the advantages of SDTT more accessible. After carefully considering the responses and other reviews, I have decided to increase my original score to reflect my updated assessment of the paper's contributions.

---

> > > ### Author Response · Authors · 2024-11-24
> > >
> > > Dear reviewer fzmS,
> > >
> > > Thank you for taking the time to provide feedback on our manuscript. We genuinely appreciate your efforts in engaging with our work and your willingness to reconsider your initial assessment based on our discussion and rebuttal.
> > >
> > > Best regards,
> > >
> > > Authors

---

### Official Review · Reviewer_ijRT · 2024-11-03

**Soundness:** 4
**Presentation:** 3
**Contribution:** 4
**Rating:** 8
**Confidence:** 3

**Summary:**

In this work, the authors propose Self-Distillation Through Time (SDTT), a training strategy for decreasing the number of sampling steps required during discrete text diffusion. Given a pretrained discrete diffusion model, the method works by using the token distribution of the base model at later denoising timesteps as the target for a student network at an early denoising timestep. The student model is initialized to be the same as the base diffusion model. The authors demonstrate that through multiple rounds of SDTT they can sample with 32x fewer steps while maintaining quality. They also compare to standard auto-regressive decoding, finding that they can achieve the same perplexity with an 8x reduction in wall clock time during generation.

**Strengths:**

* The SDTT methodology appears to significantly improve the decoding speed of discrete diffusion models, and the authors provide evidence of this over a large set of experiments. Compressing discrete diffusion sampling to this degree seems like an important contribution that will be used by future text diffusion works.

* The paper conducts a thorough set of ablations justifying key design choices such as the particular divergence measure and the number of steps compressed per SDTT round.

* The authors demonstrate that text diffusion models can scale up to 860 million parameters and promise to release said model which is a valuable contribution for future research on text-based diffusion models.

**Weaknesses:**

* Performing multiple rounds of SDTT seems integral to the empirical success the paper has but the detail that SDTT is performed over multiple rounds seems to be introduced in the experiments section. It would help the clarity of the paper if this technique was introduced as a core part of the algorithm.

* As mentioned by the authors, previous work has explored distilling multiple diffusion steps in the continuous diffusion setting ([1], [2]). While the technique in SDTT has slight differences and is applied to the discrete diffusion setting, its similarity to [2] in particular may limit the novelty of SDTT.

**Questions:**

* It would be interesting to look at why more SDTT rounds work but distilling a larger number of steps in a single round doesn’t. Did you investigate progressively growing the distillation step size throughout training?

---

> ### Author Response · Authors · 2024-11-21
> **Authors answer to the Reviwer ijRT**
>
> We sincerely thank reviewer ijRT for their thoughtful and insightful feedback. Below, we provide our responses to the questions and comments raised.
>
> > The detail that SDTT is performed over multiple rounds seems to be introduced in the experiments section.
> >
> - Thank you for the suggestion. We now introduce iterated SDTT in the methodology section, rather than the experiments section.
>
> > As mentioned by the authors, previous work has explored distilling multiple diffusion steps in the continuous diffusion setting.
> >
> - We acknowledge that distillation techniques have been explored to accelerate sampling in continuous diffusion models, and we cite several influential works on distillation in our related work section. However, we believe our contribution is novel for two primary reasons:
>     - **First**: most continuous distillation methods rely on a deterministic mapping from noise to data, which is possible due to the Ordinary Differential Equation (ODE) formulation of the reverse diffusion process. Once the continuous model is trained, this deterministic map (e.g., with DDIM) allows training a faster model via distillation. In contrast, as noted in lines 80 and 462, no such deterministic map exists for discrete (absorbing) diffusion models. Indeed, if such a deterministic map existed, only one sample could be generated from the model, since all samples from the absorbing distribution are sequences of masked tokens.
>     - **Second**, we are not aware of previous distillation methods for discrete diffusion models. We believe that it was not clear whether distillation techniques from continuous diffusion could be adapted to discrete diffusion. Our work shows that distillation is indeed possible, and we hope that it will motivate further research in speeding-up diffusion language models. Given the simplicity of our approach, we believe that there is significant room for further exploration in this area.
>
> > It would be interesting to look at why more SDTT rounds work but distilling a larger number of steps in a single round doesn’t.
> >
> - Thank you for the suggestion. We hypothesize (line 363 of the original manuscript) that since the teacher targets are stochastic, it is too difficult for the model to match targets generated over more than two steps. As evidence, we observed that the training loss is 2-4x larger when generating the teacher targets with 4 or 8 steps.
>
> > Did you investigate progressively growing the distillation step size throughout training?
> >
> - Yes, we did investigate progressively increasing the distillation step size during training. We tried several schedules, but all attempts failed, and the loss diverged after just 30-50 training steps. We have added a note about these failed experiments in the new background section, after introducing iterated SDTT.
> - **Final note**: We noticed that the two works you referenced ([1], [2]) were not linked in your review. We assume that [2] refers to progressive distillation. We have updated the introduction and methodology to emphasize progressive distillation further and to highlight the difference regarding the existence of deterministic maps.

---

> > ### Comment · Reviewer_ijRT · 2024-11-22
> > **Final Review**
> >
> > Sorry about the references being dropped, they were:
> >
> > [1] Song, Yang, et al. "Consistency models." arXiv preprint arXiv:2303.01469 (2023).
> >
> > [2] Sauer, Axel, et al. "Adversarial diffusion distillation." European Conference on Computer Vision. Springer, Cham, 2025.
> >
> > Thank you for the detailed response to my feedback. I will keep my score and vote to have the paper accepted.

---

> > > ### Author Response · Authors · 2024-11-24
> > >
> > > Dear reviewer ijRT,
> > >
> > > Thank you for responding to our rebuttal. We appreciate your time and effort spent reviewing our work.
> > >
> > > Best regards,
> > >
> > > Authors

---

### Official Review · Reviewer_7pDo · 2024-11-04

**Soundness:** 3
**Presentation:** 2
**Contribution:** 3
**Rating:** 8
**Confidence:** 3

**Summary:**

This paper proposes self-distillation through time to speed up sampling from text diffusion models. The method trains student reconstruction models to predict ahead in time, given a teacher model. This method can be applied iteratively, using students to train even coarser students. Experiments demonstrate that the method results in achieving a given generative perplexity at fewer function evaluations.

**Strengths:**

The method is simple and intuitive. The results for generative perplexity and number of function evaluations look promising. However, wall-clock time / latency is most important.

**Weaknesses:**

The primary weakness is presentation and writing. The abstract claims that the generates tokens 8x faster than an AR baseline with KV-caching. This must be presented in a figure as early as possible (see comment 10). See comments below for more suggestions.

**Questions:**

1. Notation nit: In the discrete case, with fully-factored reverse model, $p(z_s \mid z_t) = \sum_x p(z_s \mid x)p(x \mid z_t)$ can be computed efficiently (see D3PM) and (also in the discrete, fully-factored case) $x_\theta(z_t, t) = p(x \mid z_t)$.
2. Notation nit: Equation 3 can be written as $\log p(x \mid z_t)$, rather than writing a probability as the inner-product of the mean-parameters with a one-hot vector.
3. Writing: Please move iterated SDTT to the methods section and formally describe it with one equation and a few sentences. Something like a telescoping sum of divergences. Be sure to introduce terminology like "rounds" formally and do not use both "times" and "rounds" in order to minimize terms.
4. Writing: Are figure 2 and 3 referred to in the body of the text?
5. Writing: Notation such as `dt` should be formally defined in the methods section. Could you explain what `dt` is? Is it $\Delta$? Could you also define $\Delta$? In MDLM, $\Delta$ refers to a simplex rather than time-step or increment.
6. Writing: In general, "steps" can be ambiguous and can refer to sampling steps, decoding steps, or training steps. Please be clear which one is being used, for example in the beginning of section 4.1 it should be training steps. Also, likely sampling = decoding steps.
7. Question: In this setting, is MSE roughly chi-square divergence? It's surprising that using MSE results in better generative perplexity. Any thoughts on why?
8. Experiment: Could you also report the conditional likelihoods / perplexity alongside the MAUVE scores for conditional generation? It's not completely clear what MAUVE measures [1].
9. Question: It's very interesting that distilling more than 2 steps at a time, as well as more training steps per round hurt the student (Figure 11).

10: Figure request and section 4.4: In section 4.4, the following claim is made: "We successfully reproduce the results of Deschenaux & Gulcehre (2024), which showed a 4x improvement when sampling with 32 steps, only this time, we retain the text quality because of SDTT." Please add a figure that clearly shows this. A figure with latency vs generative perplexity is extremely important for this paper. Additionally, Section 4.4 is likely the most important experimental section, and should be the first result discussed.

[1] Pimentel, Tiago, Clara Meister and Ryan Cotterell. “On the Usefulness of Embeddings, Clusters and Strings for Text Generator Evaluation.” ICLR 2023.

---

> ### Author Response · Authors · 2024-11-21
> **Authors answer to the Reviewer 7pDo**
>
> We sincerely appreciate reviewer 7pDo for their valuable and insightful feedback. Below, we respond to the questions and comments raised.
>
> > The abstract claims that the generates tokens 8x faster than an AR baseline with KV-caching. This must be presented in a figure as early as possible
> >
> - We agree with reviewer 7pDo that the speedup is significant. Following your suggestion, we have moved the latency figure on the first page (now Figure 1). We are grateful for the suggestion
>
> > Notation nit: In the discrete case, with fully-factored reverse model, $p(z_s \mid z_t) = \sum_x p(z_s \mid x)p(x \mid z_t)$ can be computed efficiently (see D3PM) and (also in the discrete, fully-factored case) $x_\theta(z_t, t) = p(x \mid z_t)$.
> >
> - We are unsure if Reviewer 7pDo is suggesting a notation change or recommending we include an additional result from previous work. Could the reviewer 7pDo clarify the notation nit so that we can update the manuscript? We are currently reaching the page limit but we will do our best to include their suggestion.
>
> > Notation nit: Equation 3 can be written as $\log p(x \mid z_t)$, rather than writing a probability as the inner-product of the mean-parameters with a one-hot vector.
> >
> - We are grateful for the suggestion, Reviewer 7pDo is correct that the two notations are indeed equivalent. We agree that the suggested notation is simpler. We chose the current formulation to align with existing notation in the diffusion literature [1](https://arxiv.org/pdf/2406.07524). This consistency helps avoid introducing additional variations, making the manuscript more accessible to readers, particularly those new to the field.
> - We can modify the notation if reviewer 7pDo still believes it is necessary after reading our reply.
>
> > Please move iterated SDTT to the methods section and formally describe it with one equation.
> >
> - We appreciate Reviewer 7pDo’s suggestion. We have added a new section on iterated SDTT in the background with a new equation (equation 7 in the updated manuscript).
>
> > Writing: Are figure 2 and 3 referred to in the body of the text?
> >
> - Yes. In the original manuscript, we refer to figure 2 on lines 246-247. We refer to figure 3 on lines 202 and 416.
>
> > Notation such as dt should be formally defined in the methods section.
> >
> - We apologize for the confusion that the notation  `dt=1/k`  might have caused. We have completely removed it from the manuscript and updated all the figures to use the number of rounds of distillation instead. We hope that the reviewer 7pDo finds the updated figures clearer.
>
> > In general, "steps" can be ambiguous and can refer to sampling steps, decoding steps, or training steps.
> >
> - We agree that the term "steps" can be ambiguous. While we considered using alternatives like "iteration," the ambiguity remained. We have reviewed the paper and clarified whether we are referring to training, decoding, or sampling steps. Decoding and sampling are used interchangeably, as is common in the LLM literature. We hope this update addresses Reviewer 7pDo’s concern. If some any doubts remain, please let us know so that we can address them.
> > Question: In this setting, is MSE roughly $\chi^2$ divergence? It's surprising that using MSE results in better generative perplexity. Any thoughts on why?
> >
> - The MSE is indeed related to the $\chi^2$ divergence, although it is different. Most notably, the $\chi^2$ divergence assigns a non-uniform weight on the error for each $x$ in the discrete sample space (here the token indices), while the MSE assigns a uniform weight. Additionally, we apply the MSE on the log-probabilities computed by the model. Computing the loss in log space was more stable than exponentiating the predictions of the model and induced better performance. Following your question, we have included a paragraph on the $\chi^2$ divergence in appendix B4.
>
> > Could you also report the conditional likelihoods / perplexity alongside the MAUVE scores for conditional generation?
> >
> - We appreciate the suggestion. Figure 17 shows the perplexity of the continuations given the same prompts used to compute MAUVE. In summary, the student distilled with the KLD variant matches GPT2 with nucleus sampling using 32 forward passes. Since we are interested in conditional generation, we evaluate the perplexity of the continuation only and ignore the predictions for the prompt.

---

> > ### Author Response · Authors · 2024-11-21
> > **Authors answer to the Reviewer 7pDo**
> >
> > > “We successfully reproduce the results of Deschenaux & Gulcehre (2024), which showed a 4x improvement when sampling with 32 steps, only this time, we retain the text quality because of SDTT.”
> > >
> > - It seems that this sentence may have caused some confusion. To clarify, we observed a similar speedup at the model scale studied by Deschenaux & Gulcehre. However, since we do not train models with 1.3B parameters, we cannot report perplexity for such models. As stated in the introduction, we trained models up to 860M parameters due to cost constraints. We reported latency improvements using untrained 1.3B model because we anticipate that larger models will soon be released by researchers with more resources, and the 1.3B scale is more aligned with real-world autoregressive model workloads. We have added latency figures for various model scales (160M, 400M, 860M, 3B, 8B) in the appendix.
> >
> > > A figure with latency vs generative perplexity is extremely important for this paper.
> > >
> > - We appreciate the suggestion. We have included an additional figures to compare the generative perplexity and the wall-time latency. Please find the latency-perplexity trade-off in figure 21.
> >
> > > Section 4.4 is likely the most important experimental section, and should be the first result discussed.
> > >
> > - We agree with reviewer 7pDo that the latency improvement are a core contribution of our work. As such, we have placed the latency figure on the first page of the manuscript. Note that previous studies, such as [2](https://arxiv.org/pdf/2406.11473), had already shown that discrete diffusion models can generate tokens faster than AR models. **However, prior models saw a significant degradation in text quality** (for example see [3](https://arxiv.org/abs/2310.16834), [1](https://arxiv.org/pdf/2406.07524), [5](https://arxiv.org/abs/2406.04329)). Our main contribution is demonstrating that it is possible to reduce the number of sampling steps while maintaining performance.

---

> > > ### Comment · Reviewer_7pDo · 2024-11-25
> > >
> > > Thanks for the response. I have one more recommendation: Replace the new Figure 1 with a chart showing the Pareto frontier of speed (latency, x-axis) versus accuracy (probably generative perplexity, y-axis) of SDTT versus AR and other baselines. This will really highlight the new accuracy achieved at the given speed.

---

> > > > ### Author Response · Authors · 2024-11-25
> > > >
> > > > We are grateful to the reviewer for engaging with us.
> > > >
> > > > > Replace the new Figure 1 with a chart showing the Pareto frontier of speed (latency, x-axis) versus accuracy (probably generative perplexity, y-axis) of SDTT versus AR and other baselines. This will really highlight the new accuracy achieved at the given speed.
> > > >
> > > > - Thank you for the suggestion. We agree that comparing the latency and accuracy is important. Following your suggestion, we have updated Figure 1 to compare the latency and performance to display the trade-offs for the current models. We will update the manuscript further after the 1.3B models are done training.

---

### Official Review · Reviewer_LQck · 2024-11-04

**Soundness:** 3
**Presentation:** 3
**Contribution:** 3
**Rating:** 6
**Confidence:** 5

**Summary:**

The paper introduces a novel distillation method for discrete diffusion models: SDTT (Self distillation through time), which significantly reduces inference steps, leading to a generation process that is up to 8 times faster than AR models with KV-caching. They ablate with different distillation alternatives. They evaluate the effectiveness of MDLMs with model sizes up to 860M parameters under unconditional generation, conditional generation, and downstream LAMBADA benchmark.

**Strengths:**

- Improved Generation Quality and Speed: The paper achieves remarkable improvements in both generation quality and efficiency, with fewer decoding steps required. This is a significant advancement for the field.
- Clarity of Writing: The paper is well-written, with clear explanations of the methodologies and results, making it accessible to a wide audience.

**Weaknesses:**

- Insufficient Validation on Generation Speed: While generation speed is a key advantage highlighted in the paper, the validation seems inadequate in Section 4.4 and Figure 2b. The experimental settings lack clarity. For instance, it's not clear if the reported 8x speedup is based on 32 steps, 1.3B, and a batch size of 8. Additionally, the paper does not clarify whether the quality (generation perplexity) is comparable between 1.3B DLM and AR models.
- Impact of Batch Size and Model Size: There is a need for further validation on how batch size affects memory usage and decoding speed for both diffusion and AR models. Similarly, the paper should explore how model size impacts generation speed.
- Effect of Flash-Attention: The paper does not address whether the advantage of DLMs over AR models persists when flash-attention is employed.
- More downstream tasks such as math reasoning are encouraged.

**Questions:**

- Does the self-distillation process affect the generation diversity? There is an ongoing discussion regarding potential model collapse when AI models are trained on recursively generated data, which may be relevant here.
- How was the decision made to set the number of rounds to 7?
- The paper focuses on discrete diffusion models and generation speed for reasoning tasks but fails to cite these related works:
(1) A Reparameterized Discrete Diffusion Model for Text Generation;
(2) Diffusion of Thoughts: Chain-of-Thought Reasoning in Diffusion Language Models.

---

> ### Author Response · Authors · 2024-11-21
> **Authors answer to the reviewer LQck**
>
> We are grateful to the reviewer  LQck for the insightful feedback. Below, we address the questions raised by the reviewer LQck.
>
> > The experimental settings lack clarity. For instance, it's not clear if the reported 8x speedup is based on 32 steps, 1.3B, and a batch size of 8.
> >
> - The reported 8x speedup is achieved with 16 sampling steps (Fig. 2b), with the model with 1.3B parameters (lines 430-431). We have updated the caption of Figure 2b (Fig. 1 in the updated manuscript) to say that the 8x speedup corresponds to the model with 1.3B parameters.
>
> > Additionally, the paper does not clarify whether the quality (generation perplexity) is comparable between 1.3B DLM and AR models.
> >
> - We could only afford to train models with up to 860M parameters with the resources available to us. We reported the sampling latency for 1.3B models because this scale is closer to autoregressive models used in production workloads. Given the increasing interest in diffusion language models, we anticipate that researchers with more resources will soon release diffusion models with billions of parameters. Therefore, we considered 1B+ models more interesting for latency evaluations. Nonetheless, following your suggestion, we included latency measurements for our small, medium and large models, and for models with 3B and 8B parameters to match the sizes of the recent Llama models. As for the 1.3B parameters model, we use untrained models for the 3B and 8B scales. In summary, the smaller models achieve greater latency gains than larger models. You can find the new results in the appendix.
>
> > There is a need for further validation on how batch size affects memory usage and decoding speed for both diffusion and AR models. Similarly, the paper should explore how model size impacts generation speed.
> >
> - We are grateful for the suggestion. We have updated the manuscript with latency measurements at all model sizes and with a varying batch size. You can find them in the appendix.
>
> > Effect of Flash-Attention: The paper does not address whether the advantage of DLMs over AR models persists when flash-attention is employed.
> >
> - We use flash attention in all latency experiments. We have updated the last sentence of section 4.4 to state it explicitly. We are grateful for your help in making our work clearer.
>
> > More downstream tasks such as math reasoning are encouraged.
> >
> - We are grateful for the suggestion. In addition to LAMBADA, we have included 6 other downstream tasks, including a multiple-choice math benchmark (mathqa). You can find the results in Table 1. We observe that distillation affects the downstream performance minimally. For example, after 7 rounds of distillation, we observe an average drop in performance of 0.31%, for the student distilled with the KLD loss, trained to sample in 8 steps.
> - Let us note that all models evaluated on these tasks are relatively weak since they are only pre-trained on internet data (WebText/OpenWebText), which is not particularly designed for performance on math reasoning benchmarks.
>
> > Does the self-distillation process affect the generation diversity? There is an ongoing discussion regarding potential model collapse when AI models are trained on recursively generated data, which may be relevant here.
> >
> - We are grateful for the very relevant question. We measure the diversity of samples using the self-BLEU score, introduced by [1](https://arxiv.org/pdf/1802.01886) and used in previous work on distillation [2](https://arxiv.org/abs/2306.13649). Less diverse samples achieve a higher self-BLEU score, and deterministic samples achieve a self-BLEU of 1. We have included a formal introduction to self-BLEU in appendix A, and reported it in figures 4a, 18a and 18b.
> - We find that SDTT minimally affects diversity (i.e. there is no collapse). We observe an increase of at most 4 in self-BLEU (2 for the KLD student), while previous work  [2](https://arxiv.org/abs/2306.13649) observed an increase of 15 or more.
>
> > How was the decision made to set the number of rounds to 7?
> >
> - The original sampling step size to generate the teacher target is 1/1024, corresponding to 1024 sampling steps to map the noise distribution to the data distribution. On each distillation round, we double the step size. Therefore, during the 7th rounds of distillation, we use a step size of 1/16 to generate the teacher targets. Because we generate the teacher targets using two sampling steps from the teacher, it means the final student is trained to generate text using 8 sampling steps. We stopped the distillation because it became too difficult for the student to match the teacher targets beyond 7 rounds.

---

> > ### Author Response · Authors · 2024-11-21
> > **Authors answer to the reviewer LQck**
> >
> > > The paper focuses on discrete diffusion models and generation speed for reasoning tasks but fails to cite these related works: (1) A Reparameterized Discrete Diffusion Model for Text Generation; (2) Diffusion of Thoughts: Chain-of-Thought Reasoning in Diffusion Language Models.
> > >
> > - We are grateful for the suggestion to cite those relevant work. We now cite them on lines 503-504 and 507. Additionally, we included new experiments on reasoning benchmarks in Table 1. You can find more details in the general reply.

---

> > > ### Comment · Reviewer_LQck · 2024-11-25
> > >
> > > Thank you for your response; I appreciate your efforts in updating the manuscript.

---

### Author Response · Authors · 2024-11-21
**General reply**

Dear reviewers,

We appreciate your insightful comments. In response to your suggestions, we have improved our manuscript and conducted additional experiments, which are now included in the updated version. For ease of review, the new text is highlighted in red. Here's a summary of the updates we've made to address the reviewers' questions:

- We specify that SDTT consists of multiple rounds in the Methods section, and updated the title of Algorithm 2 to clarify.
- We updated the label for the horizontal axis for the plots that used the `1/k` notation, as it confused the reviewers. We hope that using the number of rounds of distillation is clearer.
- We have further emphasized the relationship between continuous and discrete distillation in the introduction and methods sections. Most importantly, we further discuss the progressive nature of SDTT and describe how it differs with progressive distillation in the introduction.
- We clarify in the introduction why a deterministic sampling algorithm cannot exist for discrete diffusion models, unlike continuous diffusion models.
- We evaluate our models on six additional downstream tasks, including a multiple-choice question on math.
- We measure the diversity of generated samples using self-BLEU and observe that SDTT minimally reduces the diversity. In particular, the reduction is significantly smaller than on-policy distillation of autoregressive language models [1](https://arxiv.org/abs/2306.13649). Importantly, SDTT does not cause a collapse of the output distribution.
- We evaluate the latency on smaller and larger scales and with a different batch size. We also report the trade-off between perplexity and wall-time latency in the appendix.

These new results reinforce our claim that SDTT is an effective distillation method for diffusion language models, and makes our masked language models faster than the AR baselines.

If you have any questions about the new experiments or any others, please reach out to us.

---

> ### Author Response · Authors · 2024-11-24
>
> Dear reviewers,
>
> We sincerely appreciate your suggestions and effort in reviewing our work. As the discussion period draws to a close soon, we would like to remind you that we are more than happy to address any concern you might have. Please reach out to us if further clarifications are needed.
>
> Best regards,
>
> Authors

---

### Meta-Review · Area_Chair_6LmP · 2024-12-22

**Metareview:**

The paper presents a novel distillation method, SDTT, aimed at reducing the number of sampling steps required for discrete diffusion models, thereby improving the speed of text generation. The authors have demonstrated that their method can achieve significant speedups while maintaining or improving the quality of text generation. All reviewers vote for acceptance of this paper. The main concerns lie in that authors ignore the non-autoregressive text generation literature, which are quite related to the acceleration of text generation and authors should include them for comparison and discussion.

**Additional Comments On Reviewer Discussion:**

The authors have addressed the majority of the concerns raised by the reviewers, and the paper's contributions are significant and well-supported by empirical evidence. The revisions have improved the clarity and organization of the paper.

---

### Decision · Program_Chairs · 2025-01-22

Accept (Poster)